# *MINPP1* prevents intracellular accumulation of the chelator inositol hexakisphosphate and is mutated in Pontocerebellar Hypoplasia

Ekin Ucuncu et al.[#]

Inositol polyphosphates are vital metabolic and secondary messengers, involved in diverse cellular functions. Therefore, tight regulation of inositol polyphosphate metabolism is essential for proper cell physiology. Here, we describe an early-onset neurodegenerative syndrome caused by loss-of-function mutations in the multiple inositol-polyphosphate phosphatase 1 gene (*MINPP1*). Patients are found to have a distinct type of Pontocerebellar Hypoplasia with typical basal ganglia involvement on neuroimaging. We find that patient-derived and genome edited $MINPP1^{-/-}$ induced stem cells exhibit an inefficient neuronal differentiation combined with an increased cell death. MINPP1 deficiency results in an intracellular imbalance of the inositol polyphosphate metabolism. This metabolic defect is characterized by an accumulation of highly phosphorylated inositols, mostly inositol hexakisphosphate ($IP_6$), detected in HEK293 cells, fibroblasts, iPSCs and differentiating neurons lacking MINPP1. In mutant cells, higher $IP_6$ level is expected to be associated with an increased chelation of intracellular cations, such as iron or calcium, resulting in decreased levels of available ions. These data suggest the involvement of $IP_6$-mediated chelation on Pontocerebellar Hypoplasia disease pathology and thereby highlight the critical role of MINPP1 in the regulation of human brain development and homeostasis.

[#]A list of authors and their affiliations appears at the end of the paper.

Inositol polyphosphate (IPs) comprise an ubiquitous family of small molecule messengers controlling every aspect of cell physiology[1]. The best characterized is the calcium release factor inositol trisphosphate (I(1,4,5)P$_3$ or simply IP$_3$), a classical example of a second messenger[2], generated after receptor activation by the action of phospholipase C on the lipid phosphoinositide PIP$_2$. Each of the six hydroxyl groups of the inositol ring can be phosphorylated, and the combination of these phosphorylations generates multiple derivatives[3]. Among them, inositol hexakisphosphate (IP$_6$, or phytic acid) is the most abundant in nature. In plants, IP$_6$ accumulates in seeds, within storage vacuoles, where it could represent 1–2% of their dry weight[4]. Plant seed IP$_6$ is used as the main source of phosphate and mineral nutrients (e.g., Ca$^{2+}$, K$^+$, Fe$^{2+}$) during germination. In mammalian cells, IP$_6$ is the most abundant inositol-polyphosphate species, reaching cellular concentrations of ~15–100 µM[5]. IP$_6$ is synthesized from inositol monophosphate (IP) or from IP$_3$ by the action of several inositol phosphate kinases: IPMK (inositol-polyphosphate multikinase, also known as IPK2), IP$_3$-3K (inositol -1,4,5-trisphophate 3-kinase), ITPK1 (inositol tetrakisphosphate 1-kinase), and IPPK (inositol-pentakisphosphate 2-kinase, also known as IPK1)[1,6]. Subsequently, the fully phosphorylated ring of IP$_6$ can be further phosphorylated to generate the more polar inositol pyrophosphates such as IP$_7$[7]. While IP$_6$ anabolism is well studied, its catabolism has been less characterized. Mammalian cells dephosphorylate IP$_6$ through the action of the MINPP1 (multiple inositol-polyphosphate phosphatase 1) enzyme[8] that is able to degrade IP$_6$ to IP$_3$[9]. The analysis of mouse knockouts for the inositol kinases responsible for IP$_6$ synthesis have highlighted an important role for this pathway in controlling central nervous system development, since knockout of Itpk1 or Ipmk is embryonic lethal due to improper neural tube development[10,11]. In mammals, IP$_6$ has been directly associated with a pleiotropy of functions, including ion channel regulation, control of mRNA export, DNA repair, and membrane dynamics[1]. Furthermore, IP$_6$ is considered as a natural antioxidant since its iron-chelating property enables it to inhibit iron-catalyzed radical formation[12]. Although not yet thoroughly studied, some of the physiological roles of IP$_6$ could be related to its high affinity for polyvalent cations[13,14].

To investigate the role of IP$_6$ in mammalian physiology, many studies use IP$_6$ exogenously added to cell lines in culture, often observing antiproliferative properties[15]. These studies give little attention to the chelating property of IP$_6$: cations-IP$_6$ precipitation depletes the medium of essential ions such as calcium or iron. In addition, the physiological relevance of extracellular IP$_6$ in mammals is not established. Extracellular pools of IP$_6$ have only been demonstrated in a cestode intestinal parasite[16], and several studies suggest that dietary IP$_6$ cannot be absorbed as such through the digestive system and is absent from body fluids[17,18]. Instead, de novo synthesis of IP$_6$ occurs in all mammalian cells, including in the brain with high levels in regions such as the brainstem and striatum[17,19]. The existence of several cellular pools of IP$_6$ has been suggested[6,19,20]. However, the dynamic regulation of the endogenous intracellular pools of IP$_6$ is not fully understood, since its high cellular concentration precludes the determination of IP$_6$ pool-specific fluctuations. Therefore, the exact function(s) of IP$_6$ in cell homeostasis and mammalian development remain an area of intense investigation.

Several human diseases have been genetically associated with alterations in phosphoinositide (the lipid derivatives of inositol) metabolism[21]. However, so far, no Mendelian disorder has been shown to be caused by an imbalance in the cytosolic inositol-polyphosphate pathway, with the exception of two variants in a gene involved in the conversion of the pyrophosphates forms of inositol, associated with hearing or vision impairment[22,23].

Pontocerebellar hypoplasia (PCH) is a group of early-onset neurodegenerative disorders that includes at least 13 subtypes, based on neuropathological, clinical, and MRI criteria[24,25]. PCH is usually associated with a combination of degeneration and lack of development of the pons and the cerebellum, suggesting a prenatal onset. The genetic basis is not known for all of the cases, and preliminary data from different PCH cohorts suggest that many subtypes remain to be identified. Based on the known molecular causes, PCH often results from a defect in apparently ubiquitous cellular processes such as RNA metabolism regulation and especially tRNA synthesis (i.e., mutations in EXOSC3, TSEN54, TSEN2, TSEN34, CLP1, and RARS2). Multiple additional conditions show neurological symptoms and imaging comparable to typical PCH syndromes and are caused by defects in diverse pathways involved in mitochondrial, glycosylation or purine nucleotide metabolisms. This observation further supports the disruption of ubiquitous pathways as the unexplained basis of these neurological conditions[25].

In this study, we demonstrate that loss-of-function (LOF) mutations of the MINPP1 gene cause a specific PCH syndrome. We also show that the absence of MINPP1 leads to an abnormal accumulation of intracellular IP$_6$. Using patient-derived cells, we observe that this increase in IP$_6$ is associated with impairments in neuronal differentiation and survival. In addition, we find a deregulation of cytosolic cation (e.g., Ca$^{2+}$, Fe$^{3+}$) homeostasis when IP$_6$ accumulates inside the cells. These observations suggest that the regulation of IP$_6$ by MINPP1 is critical to preserve neuronal cation homeostasis.

## Results

**Loss-of-function mutations of the MINPP1 gene are associated with a distinct subtype of Pontocerebellar hypoplasia**. To identify additional etiological diagnoses of patients with PCH, we explored a group of 15 probands previously screened negative with a custom gene panel approach[26]. Whole-exome sequencing (WES) was then performed through trio sequencing (i.e., both parents and the proband). Among the candidate genes that were identified, the MINPP1 gene was recurrent and the most obvious candidate (Table 1 and Supplementary Note). The MINPP1 gene has not been previously associated with any Mendelian disorders. To assess how frequently MINPP1 mutations could be involved in PCH, we explored two other cohorts of pediatric cases with neurological disorders. The presence of MINPP1 mutations was investigated using a custom gene panel or WES (see "Methods"). Three additional families with MINPP1 biallelic variants were identified, all the affected being diagnosed with PCH.

In total, biallelic variants in MINPP1 were identified in eight affected children from six unrelated families (Fig. 1, Table 1, Supplementary Fig. 1). These variants include homozygous early-truncating mutations in the families CerID-30 and PCH-2712, compound heterozygous missense and frameshift variants in family CerID-11, a homozygous missense variant in the endoplasmic reticulum (ER) retention domain of the protein in the family CerID-09 and homozygous missense variants in the histidine phosphatase domain of the protein in the families TR-PCH-01 and PCH-2456 (Fig. 1b, d). These four missense variants are predicted to be disease-causing using MutationTaster and SIFT[27], and involve amino acids fully conserved across evolution (Table 1, Fig. 1c). To predict the impact of the variants on protein structure, we utilized a phytase crystal structure of D. castellii and evaluated the consequences of the missense variants involving amino acids included in the model (Supplementary Fig. 1b). Tyr53Asp variant introduces a buried charge and disrupts a hydrogen bond with the donor amino-acid Ser299. The Phe228Leu substitution breaks a buried hydrogen bond with Lys241, both amino-acid positions are close to the IP$_6$ binding site. The Arg401Gln substitution replaces a buried charged

**Table 1 MINPP1 variants identified in the different cohorts.**

| Family | Genomic coordinates (GRCh38) | Variant (transcript level) | Variant (protein level) | dbSNP | ExAC | gnomAD | SIFT | Polyphen | MutationTaster | CADD |
|---|---|---|---|---|---|---|---|---|---|---|
| CERID-30 | Chr10: g.87505138_87505139insGGGGG | NM_004897.4(MINPP1): c.223_224insGGGGG | p.E75Gfs*30 | - | - | - | - | - | - | - |
| CERID-11 | Chr10:g.87505072T>G | NM_004897.4(MINPP1): c.157T>G | p.Y53D | - | - | - | Deleterious (0) | Probably damaging (0.978) | Disease causing (prob: 1) | 32 |
| | | NM_004897.4(MINPP1): c.300del | p.K101Sfs*2 | - | - | - | - | - | - | - |
| CERID-09 | Chr10:g.87552470G>A | NM_004897.4(MINPP1): c.1456G>A | p.E486K | - | - | - | Deleterious (0) | Probably damaging (0.94) | Disease causing (prob: 1) | 26.6 |
| TR-PCH-01 | Chr10:g.87552216G>A | NM_004897.4(MINPP1): c.1202G>A | p.R401Q | rs1381093602 | - | 1 allele count (freq $3 \times 10^{-6}$) | Deleterious (0) | Probably damaging (0.987) | Disease causing (prob: 1) | 33 |
| PCH-2712 | Chr10: g.87505096_87505097insGAC | NM_004897.4(MINPP1): c.181_182insGAC | p.L61* | - | - | - | - | - | - | - |
| PCH-2456 | Chr10:g.87508380T>C | NM_004897.4(MINPP1): c.682T>C | p.F228L | rs1456945513 | - | - | Deleterious (0) | Probably damaging (0.993) | Disease causing (prob: 1) | 25.8 |

CADD score ≥25 indicates that the variant is among the 0.5% most deleterious substitutions in the protein-coding regions of the human genome.

residue with an uncharged residue and disrupts a salt bridge formed with the amino-acid Asp318. Thus, all the missense variants tested are predicted to cause structural damages with potential consequences on the enzyme activity.

The eight patients presented with almost complete absence of motor and cognitive development, progressive or congenital microcephaly, spastic tetraplegia or dystonia, and vision impairments (Table 2). For most of the patients, the first symptoms included neonatal severe axial hypotonia and epilepsy that started during the first months or years of life. Prenatal symptoms of microcephaly associated with increased thalami echogenicity were detected for the individual CerID-11, while the seven other patients presented with progressive microcephaly. For patients from the families CerID-09 and 11, the phenotype appeared to be rapidly progressive and the affected children died in their infancy or middle-childhood. Strikingly, all the affected children harbor a unique brain MRI showing a mild to severe PCH, fluid-filled posterior fossa, with dilated lateral ventricles. In addition, severe atrophy at the level of the basal ganglia or thalami often associated with typical T2 hypersignal were identified in all the patients MRI (Table 2, Fig. 1a lower panel). This imaging is distinct from other PCH syndromes and thereby defines a subtype that we propose as PCH type 14.

**MINPP1 missense PCH mutations are deleterious for protein function.** The MINPP1 enzyme is predominantly localized in the ER lumen[28]. It removes phosphate groups from inositol-polyphosphate substrates starting at position 3[29,30], with high affinity for $IP_5$ and $IP_6$[31]. Indeed, it has been described as the main mammalian phytase, or enzyme involved in $IP_6$ degradation (Fig. 2a). Despite its name, this gene does not have any paralog in the human or mouse genome. In order to determine the effect of the patient mutations on the endogenous enzyme, we obtained skin fibroblasts from patients of the CerID-30 family. MINPP1 protein was undetectable in patients' cells (Fig. 2b), supporting a complete loss of function of MINPP1 as the cause of this PCH subtype. The MINPP1 gene is widely expressed in the developing and adult mouse[8,31], and rat[9], as well as in human tissues (Supplementary Fig. 2a). This broad expression pattern suggests a general role for this enzyme in regulating inositol-polyphosphate metabolism. To investigate this role, we generated a HEK293T cell model KO for the MINPP1 gene ($MINPP1^{-/-}$ HEK293) using genome editing (Fig. 2c). $MINPP1^{-/-}$ HEK293 cells showed a 30% decrease in their growth rate after 48 h of culture, likely resulting from a proliferation defect, in the absence of significant difference in the binding of the apoptosis marker Annexin-V (Fig. 2d, Supplementary Fig. 2b, c). This defect was partially rescued by transient overexpression of WT MINPP1 ($p = 0.0029$; Fig. 2e, Supplementary Fig. 2d). Contrastingly, overexpression of two of the MINPP1 missense variants (i.e., Y53D and E486K) did not rescue growth, suggesting that these variants have a major impact on the protein function. MINPP1 has two predicted N-glycosylation sites (Fig. 1d). In order to evaluate the glycosylation status of the endogenous and over-expressed WT MINPP1 as well as the Y53D and E486K missense variants, we treated the protein extract with the PNGase enzyme[32]. A shift in the molecular weight after treatment, in all the samples, indicated that MINPP1 is indeed glycosylated without major impact of the missense variants (Supplementary Fig. 2e).

**Patient and genome-edited MINPP1 mutant neural progenitors show an impaired neuronal differentiation with increased apoptosis.** To investigate the mechanism at the origin of the neurological symptoms of MINPP1 patients, we derived induced pluripotent stem cells (iPSCs) from patient CerID-30-2

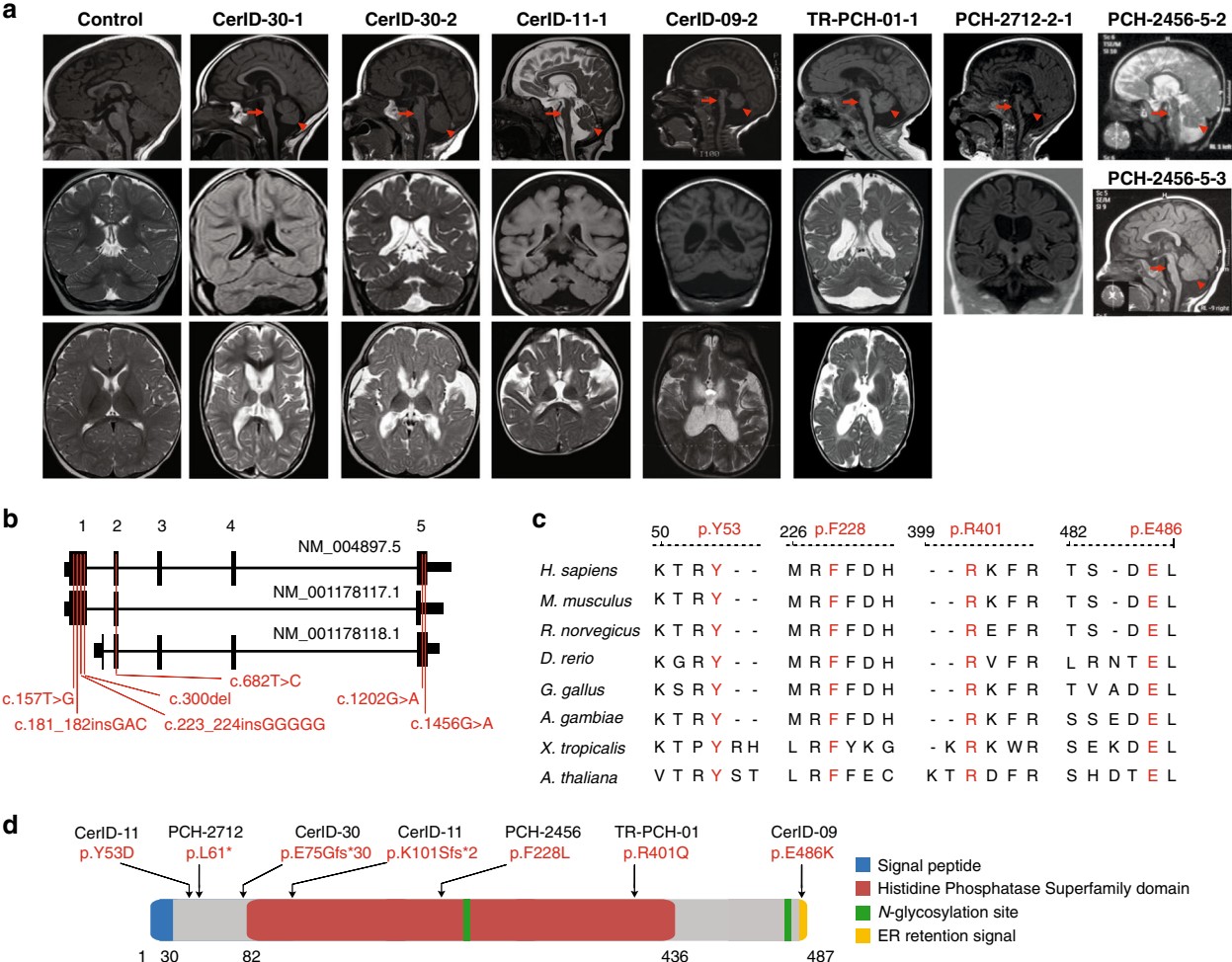

**Fig. 1 Biallelic mutations in *MINPP1* cause a distinct PCH phenotype. a** Midline sagittal (top), coronal (middle), and axial (bottom) brain MRIs of control and patients from families CerID-30, CerID-11, CerID-09, and TR-PCH-01, respectively. Only sagittal (top) and coronal (middle) brain MRIs were available for the patient from the family PCH-2712 and sagittal brain MRI for the patients from PCH-2456 (top and middle). Sagittal MRIs show variable degree of brainstem (arrow) and cerebellar atrophy/hypoplasia (arrowhead). **b** Schematic representation of the *MINPP1* transcripts: NM_004897.5, NM_001178117.1, and NM_001178118.1, respectively. Exon numbers for the longest isoform NM_004897.5 are indicated above the schematic representation. Mutations are shown relative to their cDNA (NM_004897.5) position. **c** Multiple-sequence alignment of MINPP1 from different species. Variant amino-acid residues p.Y53, p.F228, p.R401, and p.E486 are evolutionarily conserved. **d** Linear schematic representation of MINPP1, showing the position of mutations with respect to predicted protein domains. Endoplasmic reticulum (ER).

(Supplementary Fig. 3a–e). In order to assess a contribution of the genetic background or other factors to the phenotype, we also generated *MINPP1*$^{-/-}$ iPSCs in isogenic background (Supplementary Fig. 3a–e). Surprisingly, a dual SMAD inhibition-based neural induction protocol[33,34], did not allow the generation of viable neural progenitor cells for both *MINPP1* mutant lines. Differentiation of patient-derived iPSCs systematically generated mixed cell populations with undefined HNK1 negative cells (Supplementary Fig. 3h). These observations suggest a critical role for MINPP1 during neuroectodermal induction in vitro, and led us to use a different protocol that preserved neural rosette environment, using only noggin as a SMAD inhibitor[35] (Supplementary Fig. 3g). In these conditions, control cells efficiently differentiated toward TUJ1$^+$ neurons after 14 days (Fig. 3a). In contrast, *MINPP1* mutant lines showed a significant 53% decrease in TUJ1$^+$ post-mitotic cells mirrored by a significant 2.9-fold increase in the number of PAX6$^+$ neural progenitors (Fig. 3a, b). To test the presence of a neurodegenerative phenotype in our *MINPP1* mutant cells, we assessed the apoptosis level using a TUNEL assay. We observed an increase in apoptosis at day 10 of neuronal differentiation that became more significant at day 14

with a 1.9-fold increase in the number of TUNEL$^+$ cells among the *MINPP1* mutant differentiating neurons, respectively (see Fig. 3c, d). Interestingly, we observed no significant difference in apoptosis levels among the undifferentiated iPSCs lines (Fig. 3d). These results suggest that the differentiation of neural progenitors into neurons is specifically vulnerable to this inositol phosphate metabolism defect.

**Inositol-polyphosphate metabolism is altered in HEK293 cells, iPSCs, and differentiating neurons mutated for *MINPP1*.** The localization of human MINPP1 into the ER, and the demonstration that its drosophila homolog (i.e., mipp1) is anchored to the plasma membrane outside of the cell[36], prompted us to investigate the presence of phytase activity in conditioned media from control and *MINPP1* mutant HEK293 cells. In conditioned medium from control cells, exogenously added IP$_6$ was substantially processed after 2 h, and completely degraded after four hours (Supplementary Fig. 4a). Conversely, although partially degraded, IP$_6$ is still detectable after six hours of incubation in *MINPP1* mutant conditioned media. This result suggests that

**Table 2 Clinical information of the patients with variants in *MINPP1*.**

| Family | CerID-30 | CerID-30 | CerID-09 | CerID-11 | TR-PCH-01 | PCH-2712 | PCH-2456 | PCH-2456 |
|---|---|---|---|---|---|---|---|---|
| Subject | CerID-30-1 | CerID-30-2 | CerID-09-2 | CerID-11-1 | TR-PCH-01-1 | 2712-2-1 | 2456-5-2 | 2456-5-3 |
| Gender F/M | F | F | M | F | M | F | M | F |
| Origin (country) | Tunisia | Tunisia | Tunisia | France | France | Turkey | Egypt | Egypt |
| Documented consanguinity | No | No | Yes | No | Yes | Yes | Yes | Yes |
| Age at last follow-up | 12 years | 9 years | 11 years | 18 months | 12 years | 2y10m | 4.5 years | 1.5 years |
| Weight at birth (kg) | 3.2 (M) | 4.2 (+2 SD) | 3.2 (M) | 2.4 (−2SD) | 3.710 | 3.0 (M) | 3.4 (M) | 3 (M) |
| Weight at last examination (kg) | 30 (−1.5 SD) | 23.4 (−0.5 SD) | 16.9 (−4SD) | 10 (−1SD) | 36 | 10.5 (−2SD) | 10.5 (−3SD) | 8.5 (−3SD) |
| Length at birth (cm) | 50 (M) | 51 (M) | 50.5 (M) | 43.5 (−3SD) | 50 | NA | 49.2 (M) | 47.5 (−1SD) |
| HC at birth (cm) | 34 (M) | 35 (+1SD) | 35 (M) | 30.6 (−3SD) | 36 | 33 (−1SD) | 34 (M) | 32 (−1SD) |
| HC at last examination (cm, age) | 48 cm (8 years; −2SD) | 47 cm (4 years; −2SD) | 47 cm (11 years; −5SD) | 42 cm (18 months; −4SD) | −2,55 (18 months) | 41 cm (−5SD) | 44.5 cm (−5SD) | 42 (−3SD) |
| Onset | Perinatal | Perinatal | Perinatal | Prenatal | Perinatal | Perinatal | Perinatal | Perinatal |
| Progressive encephalopathy | No | No | Yes | Yes | No | Yes | Yes | Yes |
| Posture: bedridden (B), sitting (S), walking (W) | B | B | B | B | B | B | B | B |
| **Development** | | | | | | | | |
| Gross motor (normal/delayed/absent) | Absent | Absent | Absent | Absent | Absent | Absent | Absent | Absent |
| Fine motor (normal/delayed/absent) | Absent | Absent | Absent | Absent | Absent | Absent | Absent | Absent |
| Language (normal/delayed/absent) | Absent | Absent | Absent | Absent | Delayed | Absent | Absent | Absent |
| Social (normal/delayed/absent) | Absent | Absent | Absent | Absent | | | | |
| **Seizures** | | | | | | | | |
| Epileptic seizures | + | + | + | − | + | + | + | + |
| Onset | 7 years | 4 years | 5 years | − | 1 day | 2 months | 3 months | 5 months |
| **Neurological Findings** | | | | | | | | |
| Axial hypotonia | + | + | + | + | + | + | + | + |
| Distal hypertonia | + | + | + | + | + | + | + | + |
| Pyramidal signs | Spastic tetraplegia | Spastic tetraplegia | Spastic tetraplegia | Spastic tetraplegia | Spastic tetraplegia | Spastic tetraplegia | Spastic tetraplegia | Spastic tetraplegia |
| Extra-pyramidal signs | + | + | + | + | − | + | + | + |
| **Ophthalmological findings** | | | | | | | | |
| Nystagmus | + | + | + | + | + | bilateral + | + | + |
| Abnormal ocular movement | + | + | + | + | + | NA | + | + |
| Optic atrophy | + | + | + | − | − | − | − | − |
| VEP/ERG | NA | ERG | VEP | ERG/VEP | NA | abnormal VEP, left eye | abnormal VEP | − |
| Others | Ptosis/cataract | Ptosis/cataract | Blindness | Ptosis | Cataract | Ptosis, Cataract | couldn't follow object | − |
| **Investigations** | | | | | | | | |
| Metabolic | Normal | Normal | Normal | Normal | Normal | Normal | Normal | Normal |
| EEG | Abnormal | Normal | Normal | Normal | Normal | Abnormal | Abnormal | Abnormal |
| **MRI** | | | | | | | | |
| Age at last investigation (years (y); months (m); days (d)) | 2y7m | 8m2d | 1y11m | 8m23d | 4m | 6y8m | 2.5y | 1.5y |
| Cerebellum | Hypoplasia | Hypoplasia | Hypoplasia | Hypoplasia | Hypoplasia | Hypoplasia | Hypoplasia | Hypoplasia |
| Pons | Atrophy | Atrophy | Atrophy | Atrophy | Hypoplasia | Atrophy | Atrophy | Atrophy |
| Cerebral cortex atrophy | Atrophy | Atrophy | Atrophy | Atrophy | − | Atrophy | atrophy | atrophy |
| Ventricles | Enlarged | Enlarged | Enlarged | Enlarged | Enlarged | Enlarged | enlarged | enlarged |

**Table 2 (continued)**

| Family | CerID-30 | CerID-30 | CerID-09 | CerID-11 | TR-PCH-01 | PCH-2712 | PCH-2456 | PCH-2456 |
|---|---|---|---|---|---|---|---|---|
| Corpus callosum | – | – | Thin | NA | – | thin | Thin | Thin |
| Basal Ganglia hypoplasia/atrophy | + | + | + | + | thalamic atrophy | + | + | + |
| Basal ganglia T2 hypersignal | + | + | + | + | thalami | + | – | – |
| Spectroscopy | No lactate abnormality | NA | NA | NA | NA | No lactate abnormality | NA | NA |
| Others (WM defect) | WM periventricular atrophy | WM periventricular atrophy | NA | NA | NA | WM periventricular atrophy | – | – |
| Other symptoms | | | | | | | | |
| Respiratory tract congestion | No | No | Yes | No | Yes | No | Yes | No |
| Apnoea | No | No | No | No | No | No | Yes | No |
| Impaired swallowing | Yes | Yes | Yes | Yes | Yes | Yes | Yes | Yes |
| Joint stiffness | Yes | Yes | Yes | Yes | NA | Yes | Yes | No |
| Skeletal deformities | Scoliosis | Scoliosis | Scoliosis | Scoliosis | Scoliosis | Scoliosis | scoliosis | No |
| Cardiovascular findings | No | No | NA | NA | No | No | No | No |
| Urine screening | Normal | Normal | NA | Normal | Normal | recurrent urinary infection | Normal | Normal |
| Peripheral blood smear | NA | NA | NA | Thrombocytosis | Normal | NA | NA | NA |
| Muscle wasting | Yes | Yes | Yes | Yes | Yes | Yes | Yes | No |
| Facial dysmorphism | Yes | Yes | No | No | Yes | Micrognathia | Non-specific small head, prominent nose, low set ear | Non-specific small head, prominent nose, low set ears |
| Other | NA | NA | Mild Normochromic, normocytic anaemia | Abnormal fat distribution | NA | BAEP: no response | – | – |

*EEG electroencephalogram, ERG electroretinography, HC head circumference, M median, NA not available, VEP visual evoked potential.*

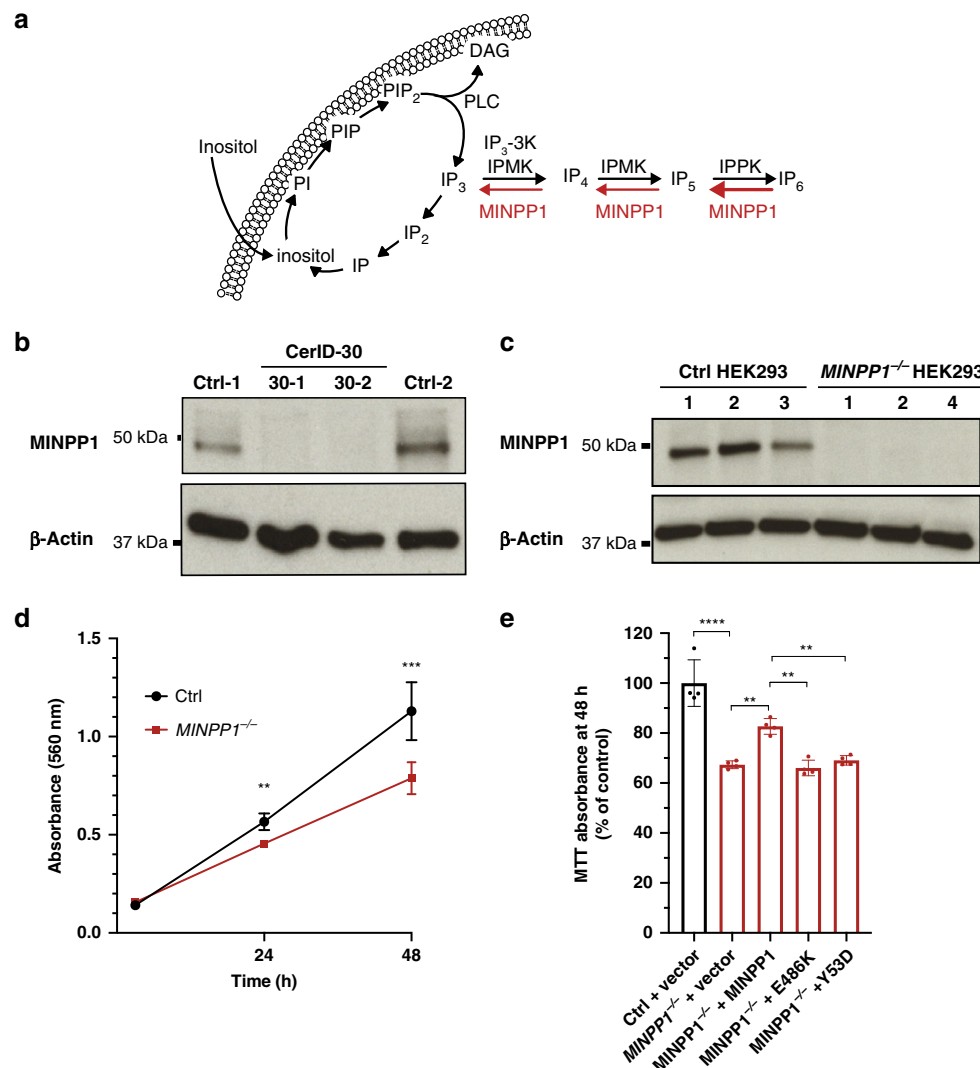

**Fig. 2 PCH-associated mutations of *MINPP1* are deleterious for protein function. a** Schematic representation of inositol phosphate cycle. *myo*-inositol (Inositol); phosphatidylinositol (PI); phosphatidylinositol phosphate (PIP); phosphatidylinositol 4,5-bisphosphate (PIP$_2$); diacylglycerol (DAG); phospholipase C (PLC); inositol phosphate (IP); inositol bisphosphate (IP$_2$); inositol 1,4,5-trisphosphate (IP$_3$); inositol 1,3,4,5-tetrakisphosphate (IP$_4$); inositol pentakisphosphate (IP$_5$); inositol hexakisphosphate (IP$_6$); inositol-polyphosphate multikinase (IPMK); I(1,4,5)P$_3$ 3-Kinase (IP$_3$-3K); inositol-pentakisphosphate 2-kinase (IPPK); multiple inositol-polyphosphate phosphatase 1 (MINPP1). **b, c** Western blot analysis of MINPP1 level in patient fibroblasts and HEK293 cells with β-Actin shown as loading control. Patient fibroblasts CerID-30-1 and CerID-30-2 (**b**) and *MINPP1*$^{-/-}$ HEK293 clones (**c**) show absent MINPP1. The uncropped blots are provided as a source data file and are representative of two independent experiments. **d** Assessment of cell proliferation by MTT assay. For each clone, MTT absorbance was measured 3, 24, and 48 h post-seeding. Values represent the mean ± s.d. of triplicate determinations from four replicates ($n = 4$, two-tailed student's *t*-test, **$p \leq 0.01$ and ***$p \leq 0.001$. At 24 h, Ctrl vs. *MINPP1*$^{-/-}$: $p = 0.0018$; at 48 h Ctrl vs. *MINPP1*$^{-/-}$: $p = 0.0009$). **e** *MINPP1*$^{-/-}$ HEK293 cells were transiently transfected with plasmids encoding empty vector, wild type, Y53D or E486K variant MINPP1. To assess the cell proliferation rate, MTT assay was performed 48 h post-nucleofection. The data are presented as mean percentage relative to control (Ctrl) ± s.d., with triplicate determinations from four replicates. The normalization was done with 3 h MTT assay data ($n = 4$, one-way ANOVA, Tukey's post hoc test, **$p \leq 0.01$ and ****$p \leq 0.0001$. Ctrl vs. *MINPP1*$^{-/-}$: $p < 0.0001$; *MINPP1*$^{-/-}$ vs. *MINPP1*$^{-/-}$ + MINPP1: $p = 0.0029$; *MINPP1*$^{-/-}$ + MINPP1 vs. *MINPP1*$^{-/-}$ + E486K: $p = 0.0014$; *MINPP1*$^{-/-}$ + MINPP1 vs. *MINPP1*$^{-/-}$ + Y53D: $p = 0.0080$). For (**d**) and (**e**), source data are provided as a source data file.

MINPP1 accounts for the main secreted phytase activity of HEK293 cells.

To explore precisely a disruption in inositol phosphate metabolism, and to better address the role of MINPP1 in this metabolic pathway, we used tritium inositol (*myo*-[³H]-inositol) metabolic labeling of cultured cells, and analyzed inositol derivatives with SAX-HPLC (strong anion-exchange high-performance liquid chromatography) as previously described[37,38]. We applied this method to HEK293 cells, skin fibroblasts, and iPSCs before or during neuronal differentiation at day 10

(referred to as day-10 differentiating neurons) from control and *MINPP1* mutant cell lines. Exogenously added [³H]-inositol is imported into the cytosol and converted into phosphoinositide lipids before processing into inositol phosphates (IPs) (Fig. 2a). As expected, after 3 days of [³H]-inositol labeling, IP$_6$ was detected as the most, or the second most abundant intracellular inositol derivative in control cell lines (black hollow bars in Fig. 4), but was absent in the cell culture media (Fig. 4, Supplementary Fig. 4b). In all the cell models studied, the disruption of MINPP1 enzyme activity had a strong impact on

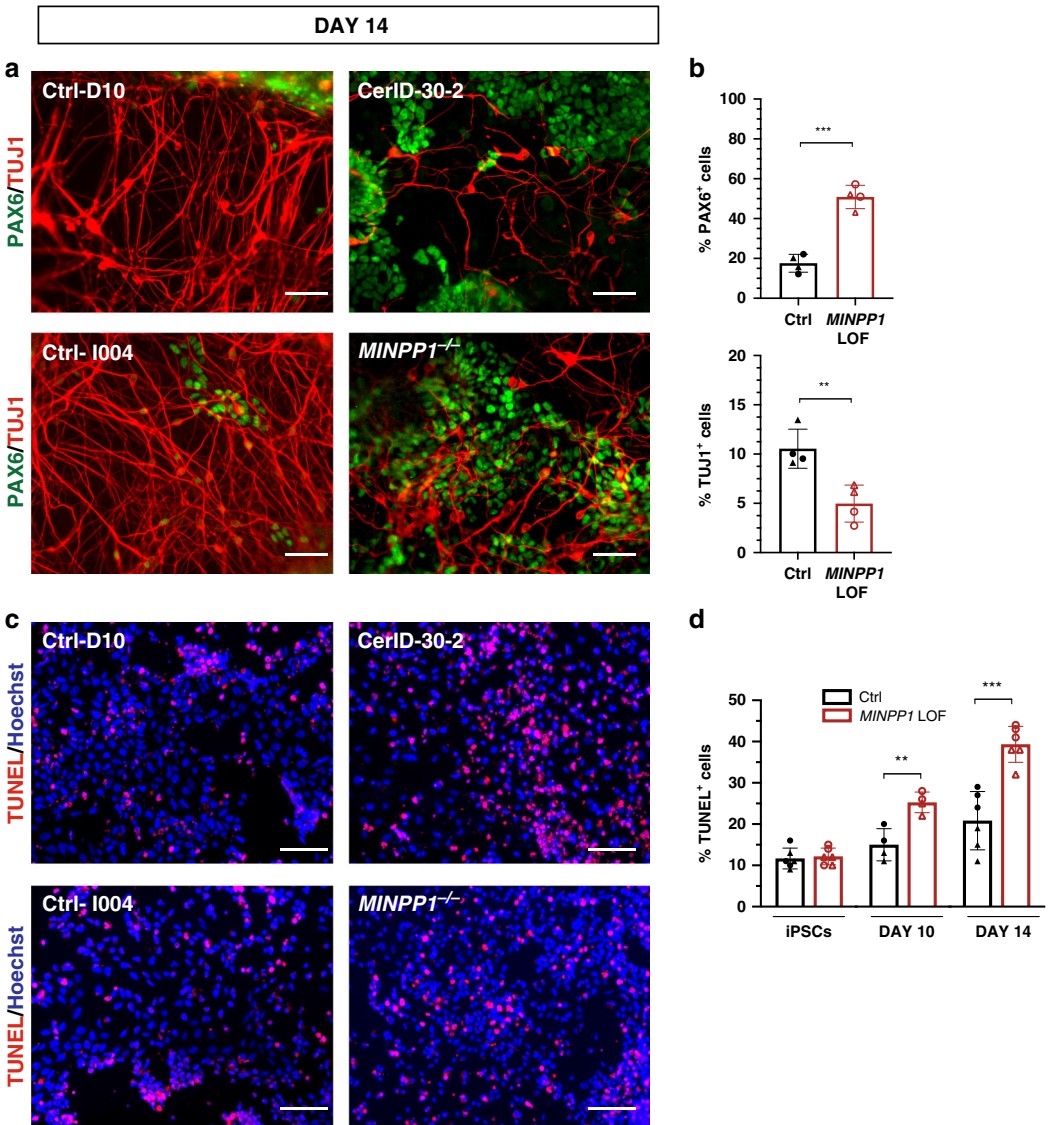

**Fig. 3 MINPP1 loss causes an early neuronal differentiation defect combined with an increase in apoptosis. a** Control (Ctrl-D10 (filled circle) and Ctrl-I004 (filled triangle)) and *MINPP1* LOF (patient-derived (CerID-30-2 (blank circle) and *MINPP1*$^{-/-}$ (blank triangle)) iPSCs were differentiated toward neuronal lineage for 14 days. Representative images of the differentiated cells stained with early neuronal marker TUJ1 and neural progenitor marker PAX6. Hoechst was used as a nuclear stain. **b** Quantitative analysis of the immunofluorescence data. **c** Representative images of TUNEL staining during neuronal differentiation at day 14. **d** Quantification of the TUNEL assay during neuronal differentiation (iPSC, day 10 and day 14). All scale bars correspond to 50 μm. (For (**b**, **c**), $n = 4$, for (**d**), $n = 6$ for iPSCs and day 14, $n = 4$ for day 10. Duplicate/triplicate analysis of two independent experiments. Two-tailed student's $t$-test, **$*p \leq 0.01$ and ***$p \leq 0.001$. For (**b**) upper panel, Ctrl vs. *MINPP1* LOF: $p = 0.0001$; for (**b**) lower panel, Ctrl vs. *MINPP1* LOF: $p = 0.0066$; for day 10 data (**d**), Ctrl vs. *MINPP1* LOF: $p = 0.0045$ and for day 14 data (**d**), Ctrl vs. *MINPP1* LOF: $p = 0.0003$. The data are presented as mean percentage values ± s.d. For (**b**) and (**d**), source data are provided as a source data file.

intracellular IPs profile when compared with their respective controls. The investigation of *MINPP1*$^{-/-}$ HEK293 cells revealed a ~3-fold increase in IP$_6$ level, as well as an increase in IP$_5$ levels, and surprisingly a severe decrease in IP and IP$_2$ levels (Fig. 4a). Differences in the same direction, although to a lesser extent, were detected in patient fibroblasts (Fig. 4b) suggesting cell-type-specific differences. Indeed, in iPSCs, IP$_6$ levels showed a significant 1.6-fold increase in patient-derived and *MINPP1* KO iPSCs, also associated with an increase in IP$_5$ levels (Fig. 4c). Finally, we studied differentiating neurons at day 10, a time point when differences in the TUJ1/PAX6 cell populations are not detected (Supplementary Fig. 3f). These data revealed significant 1.64-fold increases in IP$_6$ levels in *MINPP1* mutant differentiating cells, respectively, with a trend or significant decrease in IP$_2$ levels (Fig. 4d). Interestingly, the defects in IP$_6$ and IP$_2$ levels in

*MINPP1*$^{-/-}$ HEK293 cells were fully rescued, considering a ~70% transfection efficiency, by the transient overexpression of WT MINPP1 (Supplementary Fig. 4c, d). We also performed radiolabeling experiments on *MINPP1*$^{-/-}$ HEK cells with Y53D and E486K MINPP1 variants. The Y53D variant had no impact on the IP$_6$ and IP$_2$ levels, indicating the clear loss of enzyme activity (Supplementary Fig. 4c, d). However, the E486K variant did not affect the enzyme activity in this overexpression system, with apparently unchanged levels of IP$_6$ and IP$_2$. This result is in line with the previously demonstrated preserved enzyme activity in the absence of the ER retention signal[39]. Therefore, and given its inability to rescue the *MINPP1*$^{-/-}$ HEK293 cells proliferation, this variant could impact a critical regulation of MINPP1 protein.

In all the cell models tested, including HEK293 cells, patients' fibroblasts, and undifferentiated and differentiated iPSCs, a

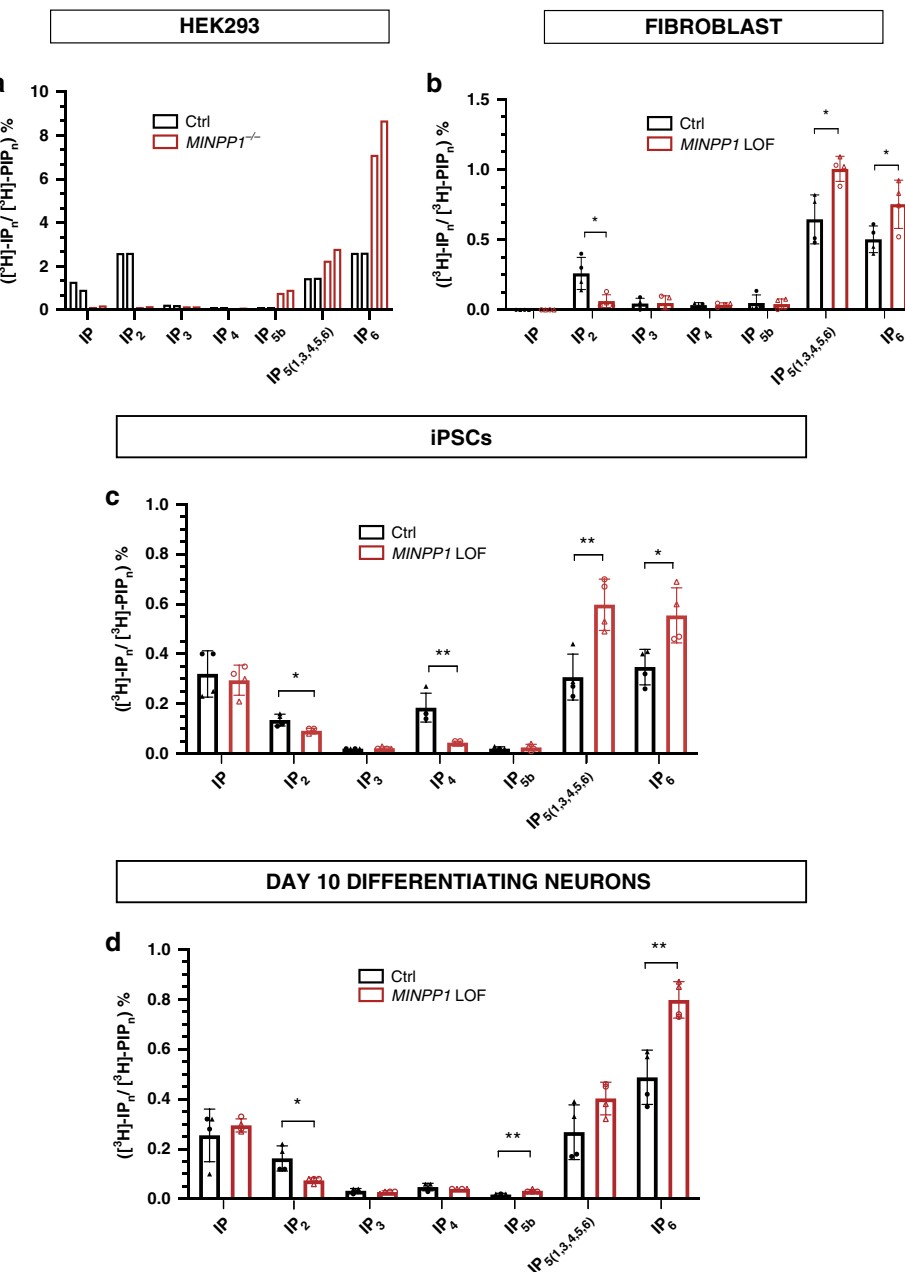

**Fig. 4 MINPP1 absence leads to disruption in inositol phosphates metabolism. a–d** SAX-HPLC analysis of inositol phosphate levels in $MINPP1^{-/-}$ HEK293 cells (**a**), patient fibroblasts (CerID-30-1 and CerID-30-2) (**b**), $MINPP1$ LOF (patient-derived (CerID-30-2 (blank circle) and $MINPP1^{-/-}$ (blank triangle)) iPSCs (**c**), and their day-10 differentiating neuron counterparts (**d**). The peaks ($[^3H]$-$IP_n$) were identified based on comparison to standards. $[^3H]$-$IP_n$ levels are presented as percentage of total radioactivity in the inositol-lipid fraction ($[^3H]$-$PIP_n$). All error bars represent standard deviation (s.d.). (For (**a**), $n = 2$ and both experiments are represented. For (**b**–**d**), $n = 4$, two-tailed student's $t$-test, $*p \leq 0.05$ and $**p \leq 0.01$). $IP_n$, inositol phosphates; $PIP_n$, phosphatidylinositol phosphates. Source data are provided as a source data file.

comparable imbalance of IPs levels were observed, where increase in the amounts of higher inositol-polyphosphate derivatives $IP_5$ and $IP_6$ were associated with a decrease in lower-phosphorylated $IP_2$ and IP species (Fig. 4). Differences observed between the various cell models tested are likely to be caused by cell-type differences and potentially also by the genetic background. Nevertheless, these observations clearly demonstrate the critical role played by MINPP1 in cellular inositol-polyphosphate homeostasis, with the conversion of higher to lower IPs. The most robust finding was that $IP_6$ is systematically increased in $MINPP1$ mutant cells compared to controls. Altogether, these observations exclude a major contribution of extracellular higher-

phosphorylated IPs to this metabolic defect, but highlight an unappreciated role for $MINPP1$ in the regulation of the intracellular pool of de novo synthesized $IP_6$, the most abundant inositol derivative.

**$IP_6$ accumulation can deplete free iron in presence of high iron condition.** Considering the strong impact of $MINPP1$ mutations on cellular $IP_6$ levels, and the known chelator properties of this molecule, we hypothesized that MINPP1 defects can have consequences for intracellular cations homeostasis. An intracellular accumulation of $IP_6$ could theoretically lead to the accumulation

of chelated cations inside the cell, potentially reducing the pool of free cations. At physiological pH, $IP_6$ has a strong binding affinity to iron[14], therefore we evaluated the ability of HEK293 cells to store iron, in low iron (−FAC) or high iron (provided with ferric ammonium citrate; +FAC) conditions. Then, we used a colorimetric ferrozine-based assay with a $HCl/KmnO_4$ pretreatment step that separates iron from its binding molecules to measure total intracellular iron[40,41]. After 2 days of incubation with FAC, we observed a significant 1.5-fold increase in the total iron content in $MINPP1^{−/−}$ HEK293 cells, under high iron conditions compared to control (Fig. 5a). Although based on non-physiological iron conditions, this observation suggests that $IP_6$ could play a role in the regulation of metal ion cellular storage such as iron. To investigate a potential increase of iron chelation affecting the free iron cellular pool, we measured the cellular free $Fe^{2+/3+}$ content using standard cell lysis and colorimetric assay. We detected a 58% depletion in total free iron levels in $MINPP1$ mutant cells under high iron conditions (Fig. 5b). Interestingly, this depletion was mainly contributed by a decrease in $Fe^{3+}$ levels (Fig. 5c, d). $IP_6$, the major IPs accumulating in mutant HEK293 cells, is known to have higher affinity for $Fe^{3+}$ versus $Fe^{2+}$ [42,43]. These data are consistent with a massive accumulation of complexed iron in the absence of the MINPP1 enzyme, in the presence of high iron, and suggest the potential involvement of $IP_6$-mediated abnormal cation homeostasis as the underlying disease mechanism.

**$IP_6$ accumulation causes cytosolic calcium depletion in mutant HEK293 cells and primary mouse neural progenitors.** To further explore the involvement of the disruption of cellular cations homeostasis in PCH, we evaluated the intracellular free $Ca^{2+}$ levels in the absence of MINPP1, using the FLUO-4-AM calcium-binding indicator. Strikingly, we found a significant, close to 50% depletion of free basal $Ca^{2+}$ levels in $MINPP1^{−/−}$ HEK293 cells, compared to control cell line (Fig. 5e). To further validate the involvement of such calcium depletion in the neurological phenotype, we generated a CRISPR/Cas9-mediated $Minpp1^{−/−}$ mouse model (Supplementary Fig. 5a–f). $Minpp1$ KO mice were fertile and are born at Mendelian ratio (Supplementary Table 1) as described in a previously generated $Minpp1$ KO mouse model[31]. Brain histology did not identify major differences in cerebellar (Supplementary Fig. 5d) or cerebral cortex architecture (Supplementary Fig. 5e). However, we identified a mild but significant ~10% decrease in the brain weight associated with a reduced cortical thickness in homozygous mutant mice at P21 (Supplementary Fig. 5c, e, f). This observation suggests the presence of an evolutionary conserved requirement for MINPP1 activity in mammalian brain development. To identify a potential cause for this cortical phenotype, we isolated neural progenitors at E14.5 and measured the intracellular free $Ca^{2+}$ levels. Surprisingly, we observed a significant 33.6% decrease in intracellular free $Ca^{2+}$ levels in the $Minpp1^{−/−}$ mouse cells when compared to wild-type neural progenitors ($p = 0.007$; Fig. 5f).

To assess the effects of $IP_6$ accumulation on calcium signaling, we studied the caffeine-sensitive ER calcium release in the $MINPP1^{−/−}$ HEK293 cells[44]. In response to 10 mM caffeine, we observed a significant peak in the cytosolic $Ca^{2+}$ levels within a minute in the control cells. However, we could only detect a slight increase with a sustained plateau in $MINPP1^{−/−}$ HEK293 cells, indicating an altered response potentially caused by a decrease of $Ca^{2+}$ in intracellular ER stores (Fig. 5g). To further study the $Ca^{2+}$ mobilization in $MINPP1^{−/−}$ HEK293 cells, we treated the cells with ionomycin as it is known to initially increase the cytoplasmic calcium levels, which in turn activates calcium-induced calcium response and eventually causes $Ca^{2+}$ depletion in the ER[45]. In

response to ionomycin, the control and $MINPP1^{−/−}$ HEK293 cells exhibited an initial $Ca^{2+}$ peak with a sustained plateau. However, the relative response to ionomycin stimulation was again significantly decreased in $MINPP1^{−/−}$ cells (Fig. 5h). These calcium signaling defects were specifically rescued in a $MINPP1^{−/−}$ HEK293 line with stable expression of MINPP1 (Fig. 5g, h). Interestingly, we observed similar results in the absence of extracellular calcium (Supplementary Fig. 5g, h), for both caffeine and ionomycin, indicating that the defect is not due to the inhibition of calcium entry. Therefore, these results clearly suggest that MINPP1 absence affects the calcium levels in the cytosol as well as in the intracellular stores such as the ER. Altogether, these data support the critical role played by MINPP1 in the regulation of the intracellular IPs and available cations with strong implications for neural cell signaling and homeostasis.

## Discussion

The direct physiological role(s) played by $IP_6$ in mammals has been difficult to define, due to the technical challenges associated with its measurement, and its complex anabolism. Furthermore, the well-described role of $IP_6$ (phytate) and phytase activity in plants and bacteria had led to the thinking that extracellular $IP_6$ degradation also supplies mammalian cells with phosphate and cations. On the contrary, we demonstrate that intracellular, not extracellular, $IP_6$ influences cation homeostasis. An imbalance of IPs derivatives has not been so far directly involved in disease, and the previously investigated $Minpp1$ KO mouse did not reveal any obvious phenotype[31]. Surprisingly, we discovered that the absence of MINPP1 in humans results in a very severe early-onset neurodegenerative disorder with specific features. Patients with loss-of-function mutations in the $MINPP1$ gene present with PCH associated with typical basal ganglia or thalami involvement identified by MRI. The prenatal onset of the phenotype is obvious for patient CerID-11, which supports a critical and early role for MINPP1 in neuronal development and survival. In agreement with this, we observed that patient-derived and genome-edited iPSCs mutant for $MINPP1$ cannot be differentiated toward neural progenitors that efficiently give rise to neurons. This defect is associated with a significant increase in apoptosis levels among the $MINPP1$ mutant cells. Interestingly, this cell survival defect could recapitulate the neurodegenerative phenotype observed in the patients. Although the exact mechanisms underlying the inefficient differentiation and the apoptosis have not been identified yet, the sensitivity of human differentiating neurons to the disruption of IPs metabolism is likely to be involved[46].

While a key role for MINPP1 in the regulation of $IP_6$ cellular levels has been investigated before[31], we provide the first evidence for its critical importance on cellular physiology and human development. Our analysis of IPs profiles using metabolic labeling unambiguously identified a typical imbalance resulting from MINPP1 defect. The increase in $IP_5$ and $IP_6$ levels is consistent with a previous mouse model study[31], but our more complete assessment of the IPs metabolism imbalance also revealed alterations in lower-phosphorylated IPs. Furthermore, the consequences of this metabolic block are associated with cell-type-dependent differences in the IPs profile, such as the $IP_4$ depletion in mutant iPSCs and robust $IP_6$ accumulation in day-10 differentiating neurons mutated for $MINPP1$.

The discrepancy related to the supposed mostly cytosolic localization of $IP_6$ and the ER localization of MINPP1 remains an unsolved problem[19,29,31]. Hypothetically, the specific subcellular localization of MINPP1 prevents $IP_6$ accumulation in a specific compartment (e.g., the ER) that would have primary consequences on local cation homeostasis. We identified that MINPP1-mediated $IP_6$ regulation impacts free cations availability, as

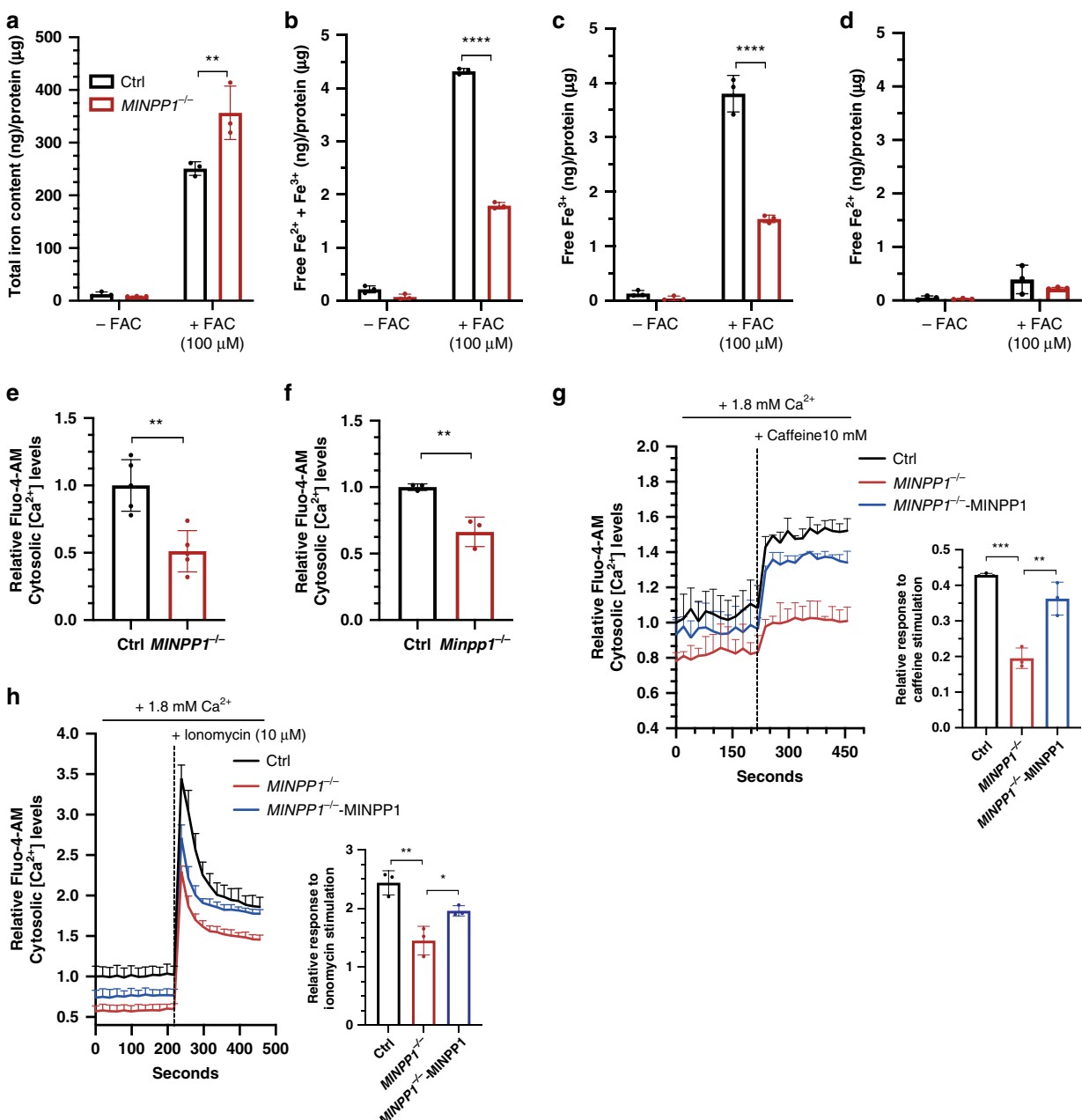

**Fig. 5 Altered iron and calcium homeostasis in the absence of MINPP1 enzyme in HEK293 and *Minpp1*$^{-/-}$ mouse neural progenitor cells.**
**a–d** Quantification of total iron content (**a**), free iron (Fe$^{2+}$ and Fe$^{3+}$) (**b**), Fe$^{3+}$ (**c**), and Fe$^{2+}$ (**d**) levels in extracts from control and *MINPP1*$^{-/-}$ HEK293 cells grown under low (−FAC) and high iron (+FAC, 100 μM) conditions. All values are normalized to the total protein concentration and represent the mean ± s.d ($n = 3$, two-way ANOVA Sidak test, **$p \leq 0.01$ and ****$p \leq 0.0001$. For (**a**), Ctrl vs. *MINPP1*$^{-/-}$: $p = 0.0023$; for (**b**), Ctrl vs. *MINPP1*$^{-/-}$: $p \leq 0.0001$, and for (**c**), Ctrl vs. *MINPP1*$^{-/-}$: $p \leq 0.0001$. **e**, **f** Relative Fluo-4-AM cytosolic Ca$^{2+}$ levels in control and *MINPP1*$^{-/-}$ HEK293 cells (**e**), wild-type (WT) and *Minpp1*$^{-/-}$ E14 mouse neural progenitors (**f**). **g**, **h** Relative Fluo-4-AM cytosolic Ca$^{2+}$ levels in control, *MINPP1*$^{-/-}$ and MINPP1-overexpression stable line in *MINPP1*$^{-/-}$ HEK293 cells (*MINPP1*$^{-/-}$-MINPP1) loaded either with 10 mM caffeine (**g**) or 10 μM ionomycin (**h**). The dotted line indicates the addition of either caffeine (**g**) or ionomycin (**h**). Relative response after caffeine or ionomycin stimulation (peak) is represented graphically (inset). For all of the calcium assay experiments, the data are normalized to cell number with MTT colorimetric assay and presented as mean values relative to control baseline fluorescence intensity control ± s.d. ((**e**) $n = 5$; (**f**) $N = 3$ mice, (**g**, **h**) $n = 3$, two-tailed student's $t$-test (**e**, **f**) and one-way ANOVA, Tukey's post hoc test (**g**, **h**), *$p \leq 0.05$, **$p \leq 0.01$, ***$p \leq 0.001$). For (**e**), Ctrl vs. *MINPP1*$^{-/-}$: $p = 0.0021$; for (**f**), Ctrl vs. *Minpp1*$^{-/-}$: $p = 0.007$; for (**g**), Ctrl vs. *MINPP1*$^{-/-}$: $p = 0.0002$; *MINPP1*$^{-/-}$ vs. *MINPP1*$^{-/-}$-MINPP1: $p = 0.0015$; for **h** Ctrl vs. *MINPP1*$^{-/-}$: $p = 0.0018$; *MINPP1*$^{-/-}$ vs. *MINPP1*$^{-/-}$-MINPP1: $p = 0.0402$. For all panels, source data are provided as a source data file.

illustrated with the altered iron content of *MINPP1*$^{-/-}$ HEK293 cells as well as the severe depletion of cytosolic calcium identified in *Minpp1*$^{-/-}$ mouse primary neural progenitors and HEK293 cells. Interestingly, the absence of MINPP1 also severely disrupts signaling based on ER calcium that could potentially be the place

of the primary defect. Calcium signaling has broad functions in neural cell physiology and brain development. Basal calcium levels influence neuronal physiology and cell survival[47–49], and calcium signaling plays a role in neural induction and differentiation[50–53]. Consequently, a disruption in calcium homeostasis

could be involved in PCH disease pathogenesis. A link between MINPP1 and calcium regulation has been suggested previously but it was through the synthesis of I(1,4,5)P$_3$[54]. Hypothetically, coupling the limitation of IP$_6$-mediated chelation of calcium with the promotion of IP$_3$ synthesis could be an efficient way for MINPP1 to regulate calcium signaling dynamics and homeostasis.

A mild or absent structural brain defect was also observed in other PCH mouse models. *AMPD2* null mutations cause PCH9 but the *Ampd2* single KO mouse is not associated with any obvious histological brain defect[55]. *CLP1* is involved in tRNA processing and mutated in PCH10, however, the *Clp1* mutant mouse showed only a mild decrease in the brain weight and volume, a phenotype overlooked before the identification of patients with a brain phenotype[56]. Differences in the phenotype of human and mouse with *MINPP1* loss-of-function mutations could be related to an increased sensitivity of the human brain development to metabolic defects, although the impact of the genetic background cannot be excluded at this point.

Disrupted cation homeostasis, including metal accumulation, is central to multiple degenerative disorders[57] such as neurodegeneration with brain iron accumulation[58,59], Parkinson's disease (manganese accumulation)[60,61], and Wilson's disease (copper accumulation)[62,63]. Basal ganglia dysfunction is usually suspected in PCH[25], however, the severe defects identified in *MINPP1* patient MRIs suggest major neurodegeneration at the level of these subcortical nuclei, a feature not typically associated with other PCH subtypes. These structures are well known to be primarily affected by metal ions accumulation and further investigation will be needed to determine how cation chelation could contribute to the disease pathogenesis. Nevertheless, our results reveal an unappreciated basic role for highly phosphorylated IPs in cellular homeostasis which is critical during neurodevelopment.

## Methods

**Patients recruitment and investigation.** The patients from families CerID-09, CerID-11, and CerID-30 included in this project were referred to the Departments of Pediatric Neurology, Genetics, Metabolism or Ophthalmology of the Necker Enfants Malades Hospital. Family TR-PCH-01 was recruited by the French Reference Centre for Cerebellar Malformations and Congenital Diseases at Trousseau Hospital. Patients from families PCH-2456 and PCH-2712 were referred for neurological or genetic assessment at the National Research Center in Cairo and Hacettepe University Medical Faculty Department of Pediatric Neurology in Turkey. Written informed consents have been obtained both from the participants and the legal representatives of the children. Details regarding sequencing, filtering, and prioritization protocols for each family are outlined in Supplementary Note.

**Protein multiple-sequence alignment.** The following protein sequences are aligned through COBALT:Multiple Alignment tool: NP_004888.2 (H. Sapiens MINPP1), NP_034929.1 (M. musculus Minpp1), NP_062136.1 (R. norvegicus Minpp1), NP_957394.1 (D. Reiro minpp1b), NP_989975.1 (G. gallus MINPP1), XP_002935472.2 (X. tropicalis minpp1), XP_313302.4 (A. gambiae AGAP003555-PA), NP_563856.1 (A. thaliana histidine acid phosphatase family protein), NP_813655 (B. thetaiotaomicron BtMinpp).

**Maintenance and culture of human-induced pluripotent stem cells (iPSCs), fibroblasts, and HEK293 cells.** All human cell culture and storage protocols were performed with approval from French Research Ministry (DC 2015-2595, 09/05/2016) and all participants provided written consent. iPSC lines Ctrl-D10 and CerID-30-2 were generated by Duke iPSC Share Resource Facility, using RNA-based reprogramming method. CerID-30-2 and Ctrl-D10 were derived from unrelated individuals. Ctrl-D10 iPSC line was derived from BJ fibroblasts of a non-disease affected male new born (ATCC cell line, CRL-2522). The control iPSC line (IMAGINi004, referred to as Ctrl-I004 in this study and described at https://hpscreg.eu/cell-line/IMAGINi004-A) was generated by Imagine Institute iPSC Platform, using non-integrating Sendai virus approach (CytoTune-2.0). The ability to make iPSC clones during reprogramming (i.e., reprogramming efficiency) was assessed for the CtrI-I004 line, from which the KO is also derived. This showed a standard ~0.3% efficacy to generate clones on vitronectin. The cloning efficiency attained during the generation of *MINPP1*$^{-/-}$ iPSC lines is ~0.05% following nucleofection and sorting. All iPSC lines were assessed for embryonic pluripotency markers OCT4, SOX2, SSEA-4, and TRA-1-81 (Supplementary Figs. 3a, b and 6a,

b). The absence of obvious chromosomal abnormalities was verified by CGH array 60 K (data not shown). The differentiation into mesoderm and endoderm lineage following embryoid body formation was assessed[64]. In addition, no obvious differences in proliferation and apoptosis were observed among all the iPSC lines. iPSCs were maintained in vitronectin-coated (10 μg/ml; 07180, STEMCELL) 3.5-cm dishes (353001, Falcon) in complete mTeSR-1 medium (STEMCELL, 85850) supplemented with 1% Penicillin-Streptomycin (PS) (15140122, Gibco) in a humidified incubator (5% CO$_2$, 37 °C). The media were replaced daily and iPSCs were mechanically passaged with ROCK inhibitor Y-27632 (10 μM; 72304, STEMCELL) every 7–8 days, with a 1:3 split ratio.

Human primary fibroblasts and HEK293T (HEK293) cells were cultured in Dulbecco's modified Eagle medium (DMEM) (11965092, Gibco) supplemented with 10% Fetal Bovine Serum (FBS) (16000044, Gibco) and 1% PS in a humidified incubator (5% CO$_2$, 37 °C).

**Generation of genome-edited *MINPP1*$^{-/-}$ HEK293 and iPSCs.** *MINPP1*$^{-/-}$ HEK293 and iPSC lines were generated via a CRISPR-Cas9 genome editing strategy, as previously described[32]. Briefly, sgRNAs targeting the first exon of MINPP1 transcript variant 1 (NM_004897.5) were designed on CRISPOR website (http://crispor.tefor.net/) and further cloned into the pSpCas9(BB)-2A-GFP plasmid (PX458, 48138, Addgene). For the generation of *MINPP1*$^{-/-}$ HEK293 clones, transfection of pSpCas9(sgRNA)-2A-GFP into HEK293 cells was performed with Lipofectamine 2000 (11668019, Invitrogen), as per manufacturer's instructions (2.5 μg of DNA per well of a 6-well plate). Two days post transfection, single GFP$^+$ HEK293 cells were sorted into 96-well plates by Fluorescence-activated cell sorting (FACS) (BD FACSAria II SORP, BD Biosciences). Indel mutations of clones were detected by Sanger sequencing, and target editing efficiency was assessed by TIDE analysis (https://tide-calculator.nki.nl/; data not shown).

For the generation of *MINPP1*$^{-/-}$ iPSCs clones, transfection of pSpCas9 (sgRNA)-2A-GFP into iPSC line (Ctrl-I004) was performed using Amaxa 4D-Nucleofector X-Unit (AAF-1002X, Lonza) according to manufacturer's instructions. Briefly, iPSCs were pre-treated with Y-27632 (10 μM) for 1 h and further dissociated into single-cell suspension with Accutase (STEMCELL, 07920). 4 × 10$^5$ iPSCs were nucleofected per 20 μl nucleocuvette strip, using P3 Primary Cell 4D X Kit S (V4XP-3032, Lonza), 1 μg DNA per reaction, and the program CA-137. After nucleofection, iPSCs were seeded on vitronectin pre-coated 12-well plates and cultured in complete mTeSR-1 medium supplemented with Y-27632 (10 μM). Two days post transfection, GFP$^+$ iPSCs were sorted by FACS and plated at low densities of 1–2 × 10$^4$ cells per 6 cm dish (Falcon, 353037) in mTeSR-1 medium supplemented with Y-27632 (10 μM). After 12–16 days, a subset of individual colonies was processed for DNA extraction using Quick-DNA/RNA Miniprep Plus Kit (D7003, Zymo) and Sanger sequenced. Target editing efficiency was assessed by TIDE analysis (https://tide-calculator.nki.nl/; data not shown). After clonal selection and expansion, no unusual chromosomal abnormalities were detected by CGH array (60 K, data not shown). Validation of *MINPP1*$^{-/-}$ iPSC clones were further assessed by the absence of MINPP1 using western blot (Supplementary Fig. 3c).

**Generation of *Minpp1*$^{-/-}$ mice.** *Minpp1*$^{-/-}$ mice were generated with the aid of LEAT platform of Imagine Institute by using a CRISPR/Cas9 system. In this study, animals were used in compliance with the French Animal Care and Use Committee from the Paris Descartes University (APAFIS#961-201506231137361). Guide RNAs (sgRNAs) targeting the first exon of the gene were designed via the CRISPOR (http://crispor.tefor.net/). C57Bl/-J female mice (4 weeks old) were super ovulated by intraperitoneal injection of 5 IU PMSG (SYNCRO-PART® PMSG 600 UI, Ceva) followed by 5 IU hCG (Chorulon 1500 UI, Intervet) at an interval of 46–48 h and mated with C57BL/6J male mice. The next day, zygotes were collected from the oviducts and exposed to hyaluronidase (H3884, Sigma-Aldrich) to remove the cumulus cells and then placed in M2 medium (M7167, Sigma-Aldrich) into a CO$_2$ incubator (5% CO$_2$, 37 °C). SgRNAs were hybridized with Cas9 (wild-type) protein and injected into the pronucleus of the C57Bl/6J zygotes. Surviving zygotes were placed in KSOM medium (MR-106-D, Merck-Millipore) and cultured overnight to two-cell stage and then transferred into the oviduct of B6CBAF1 pseudo-pregnant females. The generated knockouts were validated by Sanger sequencing combined with tide TIDE analysis (https://tide-calculator.nki.nl/; data not shown). All *Minpp1*$^{-/-}$ mice were backcrossed with C57BL/6J mice to remove potential off-targets. The *Minpp1*$^{-/-}$ offspring were identified by polymerase chain reaction (PCR) genotyping (Supplementary Fig. 5a, b). Brain histological characterization was performed with standard hematoxylin/eosin staining and analyzed with NDP.view.2.

**Generation of stable HEK293 cell lines.** HEK293 cells were transfected with either FLAG-HA-empty or FLAG-HA-MINPP1 plasmids by performing lipofection with the Lipofectamine 3000 reagent (Invitrogen, L3000015) as per manufacturer's instructions (2.5 μg of DNA per well of a 6-well plate und) under serum-free media. Six hours post transfection, the media were replaced with normal HEK293 cell culture growth media (see above). Two days post transfection, the cell culture media were replaced with the media supplemented with the selection antibiotic geneticin (0.6 mg/ml, Life, 10131027) and continued to be replaced twice

a week until the cells in the 6-well plate reach high confluence. At high confluence, the cells were passaged and continued to be cultured for 2–3 weeks with the cell culture media supplemented with the geneticin (0.6 mg/ml). The selection of clones was assessed by checking the presence of MINPP1 with western blot.

**E14.5 mouse neural progenitor cell culture.** Cerebral cortices (both halves) from E14.5 mice were dissected in Dulbecco Phosphate Balanced Salt Solution (DPBS, 14190250, Gibco). All procedures were done at room temperature, unless otherwise stated. Cortices were then incubated for 15 min at 37 °C in neurocult basal medium (STEMCELL, 05700) with Trypsin (1/100, 25300062, Gibco) and DnaseI (1/1000, 79254, Qiagen). Next, neurocult basal medium with 10% FBS was added to the mixture to deactivate the Trypsin. The mixture was then titurated with fire polished Pasteur pipettes to obtain a homogenous mixture of cells. The cells were then centrifuged for 5 min at $300 \times g$, washed twice and finally resuspended in neurocult medium with proliferation supplement (STEMCELL, 05701), endothelial growth factor (EGF) (20 ng/ml, PHG0314, Gibco) and 1% PS. Cells were then plated on poly-L-lysine (0.01 mg/ml, P6282, Sigma) and laminin (10 µg/ml; Gibco, 23017015) pre-coated dishes. Cells were passaged post 5 days of culture. All experiments were performed before passage 3.

**Differentiation of iPSCs into neural lineage.** iPSCs were differentiated into neural lineage by following a published protocol[35]. In brief, iPSCs were dissociated into small clumps with Accutase and seeded on polyornithine (0.1 mg/ml, P4957, Sigma) and laminin (4 µg/ml) pre-coated 4-well-plates (179820, Nunc) and cultured in mTSER supplemented with Y-27632 (10 µM). The day after, cell culture media was changed to N2B27 medium (1:1 mix of DMEM-F12 (31331093, Gibco) and Neural Basal Medium (1103049, Gibco)) supplemented with N2 (17502048, Gibco), B27 (17504044, Gibco), Noggin (100 ng/ml, 78060.1, STEMCELL), Y-27632 (10 µM), and 1% PS. The media were replaced every 3 days.

**Inositol labeling and SAX-HPLC analysis.** HEK293 cells and fibroblasts were plated at 10,000 cells/well on 6-well plates and radiolabeled for 5 days with 5 µCi/ml myo-[³H]-inositol (1 mCi, NET114A001, Perkin Elmer) in inositol-free DMEM medium (DML13-500, Caisson) supplemented with dialyzed FBS (Gibco, 26400-044) and 1%PS. One-quarter of the media was replaced with fresh medium containing 5 µCi/ml [³H]-inositol on day 2. For iPSCs and their day 7 neurally-differentiating counterparts, the inositol metabolic radiolabeling was performed for 3 days and inositol-free DMEM media were supplemented with either bFGF (10 ng/ml, PHG0264, Invitrogen) or Noggin (100 ng/ml, 78060.1, STEMCELL) respectively. The latter time point was chosen to analyze the inositol phosphate levels at day-10 differentiating stage, the time point corresponding to neuronal differentiation stage shortly after neural induction and before the observation of differences in the TUJ1/PAX6 cell populations between control and *MINPP1* mutant iPSCs (Supplementary Fig. 3f). For in vitro overexpression studies, the inositol metabolic radiolabeling was performed for 3 days, followed by lipofectamine-based transfection into HEK293 cells in the presence of 5 µCi/ml myo-[³H]-inositol. One day post transfection, the inositol phosphates were extracted. Extraction of inositol phosphates was performed according to published protocol[37]. Briefly, cells were incubated on ice with 1 M perchloric acid and 5 mM EDTA for 10 min followed by the collection of extracts and an overnight neutralization step with 1 M potassium carbonate. Inositol phosphates were then separated by strong anion-exchange high-performance liquid chromatography (SAX-HPLC). The peaks ([³H]-IP$_n$) were identified based on comparison to standards according to a previously published study[37]. The radioactivity within each fraction was measured with a beta counter after addition of 4 ml of scintillation fluid (Ultima-Flo AP LCS-mixture; Packard). The data are presented as percentage of total radioactivity in the inositol-lipid fraction, obtained by incubating the post-extraction cells with 0.1% NaOH and 0.1% SDS overnight at room temperature.

**Cloning of MINPP1 variants into plasmids.** The MINPP1 cDNA (transcript NM_004897.5) sequence was isolated by PCR from human brain cDNA. The PCR product was then ligated into the FLAG-HA-pcDNA3.1 vector (52535, Addgene) between XhoI and BamHI sites. FLAG-HA-pcDNA3.1 was a gift from Adam Antebi (Addgene plasmid # 52535; http://n2t.net/addgene:52535; RRID: Addgene_52535). The mutations c.157T>G and c.1456G>A were incorporated into the wild-type plasmid with QuickChange Lightning Site-Directed Mutagenesis Kit (210515, Agilent). One Shot TOP10 chemically competent *E. coli* (C404006, Invitrogen) were then transformed with the resulting plasmid and plated on ampicillin-resistant agar plates. Plates were incubated at 37 °C overnight. Colony PCR was performed on plasmids by using T7 promoter/BGH-rev primers and clones were sequenced by Sanger sequencing to check sequence reliability.

**Transfection experiments for proliferation studies using HEK293 cells.** Transfection into HEK293 cells was performed using Amaxa 4D-Nucleofector X-Unit (AAF-1002X, Lonza) according to manufacturer's instructions. In brief, $2 \times 10^5$ HEK293 cells were nucleofected per 20 µl nucleocuvette strip, using P3 Primary Cell 4D X Kit S (V4XP-3032, Lonza), 1 µg DNA per reaction, and the program CA-137.

**MTT proliferation assay.** Cells were cultured with methylthiazolyldiphenyl-tetrazolium bromide (MTT) (0.5 mg/ml, M5655, Sigma) for 1.5 h. Then, MTT and the cell culture medium were removed and 100% dimethyl sulfoxide (100 µl/well) was added to dissolve the formazan crystals. Next, the cell culture plate was left to shake for 15–30 min in the dark at room temperature, followed by transfer of the samples into flat-bottom 96-well microtiter plate (655101, Greiner). The optical density was read on a microplate reader at 560 nm (1681135, Biorad).

**Apoptosis assay.** The apoptosis is measured using the Click-iT TUNEL Alexa Fluor 594 Imaging Assay (C10246, Thermofischer) according to manufacturer's instructions. For a positive control, cells are treated with tunicamycin (10 µg/ml, 3516/10, Biotechne) for 6 h to induce apoptosis (data not shown). For quantification analysis, a minimum of 5 fields per replicate were analyzed in ImageJ. The percentage of TUNEL⁺ cells was counted relative to the total number of Hoechst⁺ cells per field which was comparable (i.e., Hoechst⁺) between replicates and conditions.

**Quantification of free intracellular Ca²⁺ levels.** Cells were plated at 25,000 cells/well on flat-bottom 96-well plates (Greiner, 655088). After 24–48 h in culture, the cells were loaded with Fluo-4-AM (5 mM, F14201, Invitrogen) diluted in assay buffer (NaCl 140 mM, Glucose 11.5 mM, KCl 5.9 mM, MgCl$_2$ 1.4 mM, NaH$_2$PO$_4$ 1.2 mM, NaHCO$_3$ 5 mM, HEPES 10 mM, pH 7.4) with or without CaCl$_2$ (1.8 mM) to a final concentration of 5 µM. After 30 min of incubation in a humidified incubator (5% CO$_2$, 37 °C), the loading medium was removed and the cells were washed twice with the assay buffer. Then, the cells were excited at 485 nm, and emission intensities were recorded at 535 nm via a fluorometers (200ProTecan and Tristar LB941) in the presence of assay buffer. The measured fluorescence intensities were then normalized to cell number via MTT proliferation assay. For Ca²⁺ stimulation experiments, Ionomycin (10 µM, I0634-1MG, Sigma) or Caffeine (10 mM, C0750, Sigma) were quickly added to the wells in the presence of assay buffer and fluorescence intensities were recorded every 30 s for a minimum of 10 min. All the Ca²⁺ stimulation experiments were performed with early batch HEK293 cells[65].

**Quantification of intracellular total and free iron levels.** The total iron content was measured with a colorimetric ferrozine-based assay according to previous studies[40,41]. Briefly, cells were cultured on 6-well plate (92006, TPP) and incubated with or without ferric ammonium citrate (FAC) (100 µM, RES20400-A702X, Sigma) in serum-free DMEM for 48 h. Then, the cells were lysed with NaOH (50 mM, 200 µl/well) and 50 µl of the lysate was kept apart for protein quantification. The remaining cell lysate was mixed with equal volumes of 10 mM HCl and iron-releasing reagent (HCl 1.4 M, KMnO$_4$ 4.5%) and incubated for 2 h at 60 °C. Once the samples reached room temperature, 30 µl of iron-detection reagent (6.5 mM ferrozine, 6.5 mM neocuproine, 2.5 M ammonium acetate, and 1 M ascorbic acid) was added to each sample. After 30 min of incubation at room temperature, the samples were transferred to flat-bottom 96-well microtiter plate (Greiner, 655101). The optical density was read on a microplate reader at 560 nm (Biorad, 1681135). FAC was used as reference standard.

The free iron levels (Fe²⁺ and Fe³⁺) were measured by using the Iron Assay Kit (Abcam, ab83366) according to manufacturer's instructions. In both iron-detection assays, the data are normalized against the protein levels, quantified via Pierce BCA Protein assay kit (23225, Thermo Scientific).

**Immunofluorescence.** Cells were fixed in cold 4% paraformaldehyde for 10 min followed by three washes of PBS and permeabilization in 0.25% PBS-Triton X-100 for 10 min. Cells were blocked for 1 h in 1% BSA and 22.52 mg/ml glycine diluted in 0.1% PBS-Tween-20 (PBST). Cells were then incubated overnight at 4 °C with the following primary antibodies diluted in 1% BSA/0.1% PBST: anti-PAX6 (1:400, 901301, BioLegend), anti-TUBULIN β-3 (1:1000, 801201, BioLegend), anti-OCT4 (1:100, sc-5279, Santa-Cruz), anti-SOX2 (1:1000, AB5603, Millipore), anti-NESTIN (1:200, N5413, Sigma). After three washes with 0.1% PBST, cells were incubated for 1 h with Alexa Fluor-coupled secondary antibodies (A21424, A-11034, Invitrogen) diluted 1:500 in 0.1% PBST. Following three washes with 0.1% PBST, cells were then incubated with Hoechst 33342 (H3570, Invitrogen) diluted 1:1000 in 0.1% PBST. After 10 min of incubation, cells were washed twice with 0.1% PBST and either kept in PBS or mounted with ProLong™ Diamond Antifade Mountant (P36965, Invitrogen). Images were acquired with CELENA-S Digital Imaging System (CS20001, Logos) and Zeiss Axioplan-2. For quantification analysis, a minimum of 10 fields per replicate were analyzed in ImageJ Java 1.8.0. The percentage of PAX6⁺, TUJ1⁺, or BrdU⁺ cells were counted relative to the total number of Hoechst⁺ cells per field which was comparable (i.e., Hoechst⁺) between replicates and conditions.

**Protein extraction and western blot analysis.** Protein extraction, quantification, separation in gel electrophoresis, and transfer were performed using standard procedure[32]. Nitrocellulose membranes were blocked either in Odyssey-TM Blocking Buffer (927-50003, LICOR) or in 5% dry milk diluted in 0.2% PBST for 1 h and further incubated overnight at 4 °C with the following primary antibodies diluted in either Odyssey-TM Blocking Buffer or 2.5% dry milk diluted in 0.2%

PBST: anti-MINPP1 (1:2000, sc-10399, Santa-Cruz), anti-β Actin (1:5000, AM4302, Invitrogen). After three washes with 0.2% PBST, the membranes were then incubated for 1 h at room temperature with HRP (1:10,000, sc-2314, sc-2020, Santa-Cruz) or IRDye-coupled (1:10,000, 925-68070, 925-32211, LICOR) secondary antibodies. After three washes with 0.2% PBST, the membranes were developed either with HRP (ChemiDoc XRS, Biorad) or Odyssey CLx imaging system (LICOR) and analyzed with Image Studio Lite Ver 5.2. Uncropped scan blots are available in the Source data file.

**Statistics**. Data were analyzed where appropriate with one-way analysis of variance (ANOVA), two-tailed student's *t*-test, and two-way ANOVA with GraphPad Prism 8 and Microsoft Excel 2016. N and n values are detailed in figure legends. All the data are presented as mean ± the standard deviation (s.d.). Statistically significant differences are indicated on the figures by $*p \leq 0.05$, $**p \leq 0.01$, $***p \leq 0.001$, or $****p \leq 0.0001$.

**Reporting summary**. Further information on research design is available in the Nature Research Reporting Summary linked to this article.

## Data availability
Patient's consent precluded the deposition of WES data in public database. These sequence data are available from the corresponding author upon reasonable request. All the other source data and uncropped blots are provided within the source data file.

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

## Acknowledgements

The project is funded by the French National Research Agency ANR-16-CE12-0005-01 (V.C.), the Fondation pour la Recherche Médicale FRM-DEQ20160334938 (L.C. and V.C.), and FRM-FDT201904007935 (E.U.). This work was also supported by State funding from the Agence Nationale de la Recherche under "Investissements d'avenir" program (ANR-10-IAHU-01), the Fondation Bettencourt Schueller, and the MSDAvenir fund (DEVO-DECODE project). A.S. and M.W. are supported by the Medical Research Council (MRC) award (MR/T028904/1). We are grateful to families of patients for their participation and Dr. B. Isidor for the clinical-follow-up. We sincerely thank S. Jaalouk, V. Meneghini, F. Petit, and A. Rotig for technical help and advice.

## Author contributions

Recruitment and evaluation of subjects: G.B., M.-T.V.-D., G.P., E.L., N.R., E.B., M.Z., M.T., F.M.S., D.M., F.G., A.M., N.B., N.B.-B., and C.F. Genetic sequencing and interpretations: N.A., V.S., C.B.-F., P.N., L.C., L.B., J.G.G., and V.C. Inositol labeling and analysis: M.W. and A.S. iPSCs generation and characterization: E.U., D.M.-C., K.R., C.B., and N.L. Mouse generation and characterization: K.R. and P.D. Neural differentiation and cell biology experiments: E.U. and K.R.

## Competing interests

The authors declare no competing interests.

## Additional information

Ekin Ucuncu[1,24], Karthyayani Rajamani[1,24], Miranda S. C. Wilson[2], Daniel Medina-Cano[1], Nami Altin[1], Pierre David[3], Giulia Barcia[1,4], Nathalie Lefort[5], Céline Banal[5], Marie-Thérèse Vasilache-Dangles[6], Gaële Pitelet[7], Elsa Lorino[8], Nathalie Rabasse[9], Eric Bieth[10], Maha S. Zaki[11], Meral Topcu[12], Fatma Mujgan Sonmez[13,14], Damir Musaev[15], Valentina Stanley[15], Christine Bole-Feysot[16], Patrick Nitschké[17], Arnold Munnich[18],

Nadia Bahi-Buisson[19], Catherine Fossoud[20], Fabienne Giuliano[21], Laurence Colleaux[1], Lydie Burglen [1,22], Joseph G. Gleeson [15], Nathalie Boddaert[23], Adolfo Saiardi [2✉] & Vincent Cantagrel [1✉]

[1]Université de Paris, Developmental Brain Disorders Laboratory, Imagine Institute, INSERM UMR 1163, F-75015 Paris, France. [2]MRC Laboratory for Molecular Cell Biology, University College London, WC1E 6BT London, UK. [3]Transgenesis Platform, Laboratoire d'Expérimentation Animale et Transgenèse (LEAT), Imagine Institute, Structure Fédérative de Recherche Necker INSERM US24/CNRS UMS3633, 75015 Paris, France. [4]Département de Génétique Médicale, AP-HP, Hôpital Necker-Enfants Malades, F-75015 Paris, France. [5]Université de Paris, iPSC Core Facility, Imagine Institute, INSERM UMR 1163, F-75015 Paris, France. [6]Département de Neurologie Pédiatrique, AP-HP, Hôpital Necker-Enfants Malades, F-75015 Paris, France. [7]Service de Neuropédiatrie, CHU Nice, 06200 Nice, France. [8]ESEAN, 44200 Nantes, Service de maladies chroniques de l'enfant, CHU Nantes, 44093 Nantes, France. [9]Service de pédiatrie, hôpital d'Antibes-Juan-les-Pins, 06600 Antibes-Juan-les-Pins, France. [10]Service de Génétique Médicale, CHU Toulouse, 31059 Toulouse, France. [11]Clinical Genetics Department, Human Genetics and Genome Research Division, National Research Centre, Cairo 12311, Egypt. [12]Department of Child Neurology, Faculty of Medicine, Hacettepe University, Ankara 06100, Turkey. [13]Guven Hospital, Child Neurology Department, Ankara, Turkey. [14]Department of Child Neurology, Faculty of Medicine, Karadeniz Technical University, Trabzon 61080, Turkey. [15]Laboratory for Pediatric Brain Diseases, Rady Children's Institute for Genomic Medicine, University of California San Diego, La Jolla, CA 92093, USA. [16]Université de Paris, Genomics Platform, Imagine Institute, INSERM UMR 1163, F-75015 Paris, France. [17]Université de Paris, Bioinformatics Core Facility, Imagine Institute, INSERM UMR 1163, F-75015 Paris, France. [18]Université de Paris, Translational Genetics Laboratory, Imagine Institute, INSERM UMR 1163, F-75015 Paris, France. [19]Université de Paris, Genetics and Development of the Cerebral Cortex Laboratory, Imagine Institute, INSERM UMR 1163, F-75015 Paris, France. [20]Centre de Référence des Troubles des Apprentissages, Hôpitaux Pédiatriques de Nice CHU-Lenval, 06200 Nice, France. [21]Service de Génétique Médicale, Centre Hospitalier Universitaire de Nice, 06202 Nice, France. [22]Centre de Référence des Malformations et Maladies Congénitales du Cervelet, Département de Génétique, AP-HP, Sorbonne Université, Hôpital Trousseau, 75012 Paris, France. [23]Département de radiologie pédiatrique, INSERM UMR 1163 and INSERM U1000, AP-HP, Hôpital Necker-Enfants Malades, F-75015 Paris, France. [24]These authors contributed equally: Ekin Ucuncu, Karthyayani Rajamani. ✉email: a.saiardi@ucl.ac.uk; vincent.cantagrel@inserm.fr

