## [Peer Review File · Nature Communications]

Reviewers' comments:

Reviewer #1 (Remarks to the Author):

The author's central hypothesis is that mutations in MINPP1 lead to alterations in IP5/IP6 levels that perturb neuronal development, leading to pontocerebellar hypoplasia (PCH) in humans. No sound mechanistic basis for the phenotype is advanced. The proposed mechanistic basis – that increased IP6/cation chelation reduces free levels of iron and calcium – arises (in the case of iron) out of poorly designed unphysiological experiments and (in the case of calcium) an oversimplified interpretation of the data, and a failure to consider very plausible but alternative interpretations. In any case, even if there were more cation chelation, it is not obvious how that could lead to the PCH condition. To some extent, these criticisms could be addressed by further experiments.

1. The human genomic evidence of a link between MINPP1 mutations to PCH (Fig. S1) is very preliminary: sequencing was performed for only 10 individuals from 3 families. Also, in family CerID-09, there is an apparent lack of sequencing data for MINPP1 in one PCH individual and 4 unaffected siblings. Such positive and negative control data would allow a more complete picture on the degree to which MINPP1 mutations associate with the PCH phenotype. Presumably such data cannot be obtained, but their absence does weaken the conclusions. To reflect the preliminary nature of these data, the title of Fig. 1 should be modified to indicate that mutations in MINPP1 are associated with the PCH phenotype, not the "cause" of it. Furthermore, in the text (page 5, 4 lines up), the authors state that MINPP1 was just one of an unspecified number of "new candidate genes that were identified." If there is a stronger association of any of these mutations with the phenotype, they should be described.

Another problem is that among just 4 PCH individuals that provide sequencing data, only 2 of these can currently be considered as providing direct support for the author's hypothesis that changes in MINPP1 catalytic activity underlie PCH: the early truncation mutant for family CerID-30, which surely destroys MINPP activity. To be fair, the PCH condition is very rare, so sample numbers are inevitably low. Nevertheless, the authors have no direct evidence that the 2 other mutations – Tyr53Ala and Glu486Lys – have any impact on MINPP1 activity and IP5/IP6 metabolism *in vivo*.

However, Tyr53Ala is not too far from the active site and could have a previously unappreciated catalytic impact; or it could indirectly alter activity by affecting protein folding. The latter should at least be studied *in silico*, perhaps using the Missense3D algorithm of the PHYRE structural prediction server. As for Glu486Lys, it lies in a functionally-distinct, ER retention tetrapeptide (1, 2), which should be noted; if the effect of that mutation is to enhance MINPP1 secretion, it could reduce the amount of cellular enzyme that has access to substrates, also raising IP5/IP6 – i.e., phenocopying the MINPP1 knockout. **This issue is so central to the author's hypothesis that they must directly assess the catalytic activities of recombinant versions of the Tyr53Ala and Glu486Lys mutants and/or the impact of their over-expression on IP5/IP6 levels in intact cells.** The apparent absence of an effect of these two mutations upon cell growth (Fig. 2E) does not provide any direct evidence as to the state of the catalytic activity.

The subject of MINPP1 secretion also needs expanding slightly, because Fig. S2D appears to show that overexpressed Tyr53Ala and Glu486Lys migrate as a doublet following Western analysis, while WT enzyme appears as a single band. This, too, deserves comment. The doublet likely reflects glycosylation as MINPP1 progresses through the secretory pathway (3, 4), consistent with the possibility (see above) that the Glu486Lys mutation is a gain-of-secretion mutant. The authors should therefore study this possibility by assessing levels and catalytic activities of the two mutants in conditioned medium – are they different from secreted WT enzyme?

If on the other hand the authors were to establish one or both of the Tyr53Ala and Glu486Lys mutations do not alter cellular IP5/IP6 levels, they may wish to conclude that they have a false-positive association with the PCH condition.

The authors should also report the results of screening appropriate databases for the incidence of each of the three mutations they have described.

2. The partial rescue of the MINPP1--induced proliferation defect by overexpression of wild-type MINPP1 is of potential interest, but the authors must also perform the positive control, i.e., determine if MINPP1 over-expression at least partially restores cellular IP5/IP6 levels. This experiment is made even more vital by a prior observation that WT MINPP1 over-expression in ER does not alter IP5/IP6 levels (5). Obtaining these metabolic data for wild-type enzyme is also a necessary control for studying the effects of the Tyr53Ala and Glu486Lys mutations upon IP5/IP6 levels (see point 1).

3. page 18: "we identified a mild but significant ~10% decrease in the brain weight associated with a reduced cortical thickness in [minpp ko] mice at P21. This observation suggests the presence of an evolutionarily conserved requirement for MINPP1 activity in mammalian brain development, despite apparent differences in the severity of the phenotype." This is rather an exaggeration; a slightly smaller brain size and a thinner cortex is not a recapitulation of a less severe PCH phenotype. Delete the phrase "despite apparent differences in the severity of the phenotype."

Some might argue that the inability of the minpp-null mice to recapitulate the human PCH condition is of detriment to the study, particularly since (see point 1), the human genetic data are themselves very preliminary. However, the central concept that MINPP1 deletion compromises neuronal differentiation is supported by data in Figs 3,4 that show patient-derived and MINPP1-null iPSCs exhibit elevated IP6 levels and impaired neuronal differentiation. And in any case, human brains are different from mouse brains. While mice brain development may be able to (largely) compensate for minpp loss, and humans cannot (a point that could be emphasized), isn't that in itself a significant and interesting observation? That being said, these data from iPSCs only validate an effect of the MINPP1-null condition, not the Tyr53Ala and Glu486Lys mutations; this point should be noted.

4. The authors construct a hypothesis concerning cation-IP6 binding that ignores complexities and uncertainties concerning both the disposition of cellular IP6 and its access to MINPP1. The authors need to be more transparent about published data that challenge the simplicity of their ideas.

(a) For example, page 3, 4-5 lines up. "[IP6 has a] cytosolic concentration of 50-100 μ M". This misquotes a statement from citation 5; the latter actually describes the range as being 15-100 μ M, and goes on to state that this range represents estimates of total cellular IP6 concentration, and, furthermore, "much of this InsP6 may not be freely soluble." More recent work has established that IP6 is a structural cofactor for certain proteins, further reducing its free levels. The author's text should be clarified to avoid it being misinterpreted as meaning free cytosolic IP6 exists at up to 100 μ M. The actual free IP6 concentration is unknown, but likely far less than 100 μ M.

(b) Page 4, 7-9 lines up. "The dynamic regulation of **the** endogenous pool of IP6 is not fully understood." [my emphasis]. This implication that there is just one cellular pool of IP6 is not correct. IP6 that acts as a structural cofactor is surely not in the same pool as cytoplasmic IP6. Also, Otto et al (6) have published good evidence IP6 exists in different metabolic pools. Does each pool have equal access to MINPP1?

(c) On page 8 the authors correctly describe cellular MINPP1 as being localized in the ER lumen. It should also be noted that it's unclear how MINPP1 accesses its substrates. That background better justifies why the authors appropriately highlight their observation that there are higher cellular

levels of MINPP1 substrates when the enzyme is knocked-out (page 14, last sentence). Nevertheless, they should note here that others have previously described this very phenomenon (7).

5. In Fig. 2A, the authors assay total cellular iron in WT and MINPP1^{-/-} cells. Under physiologically-relevant, low-iron conditions, there is no impact of the MINPP1^{-/-} genotype upon iron levels (it's hard to tell from the Y-axis scaling, but the knockout cells may actually have less iron). But there is a qualitatively different outcome when the authors incubate the cells for 48 hours with unphysiologically high extracellular iron concentration (100 μM): then, the knockout cells accumulate more iron. The reliance of this experimental result on purely non-physiological conditions (that greatly perturbs iron homeostasis (8)) renders its conclusions unreliable.

See Fig. 5C,D. There is a fundamental experimental flaw in the efforts to assay free cellular Fe²⁺ and Fe³⁺; it is not possible to obtain the data in intact cells, so the authors pursue this goal with cell extracts. Imagine the furor that would result if a journal published a paper that used the same approach to record free cytoplasmic calcium!

Just as is the case for calcium, iron is present in different cellular pools. In intact cells, iron is distributed between cytoplasm, mitochondria, lysosomes and endosomes; much cytoplasmic iron is stored within ferritin complexes (8). Very little iron is actually 'free'. Moreover (point 4), IP6 appears to be compartmentalized too. After cell lysis, this compartmentalization is largely destroyed. Under such in vitro conditions, when IP6 and iron are now free from compartmentalization, the known tight chelation of iron by IP6 binding is expected to materialize. Thus, the cell lysates that contain the most IP6 (i.e., those prepared from MINPP1^{-/-} cells) will also exhibit the least free iron. That is, the experiments described in Fig 2C,D are "doomed to succeed." The resulting in vitro data are not evidence of IP6 reducing free cytoplasmic iron levels in intact cells as a mechanism of action relevant to neurological disorders. Note also (see above), such data are obtained in the context of cell preincubation in an unphysiologically high extracellular iron concentration and hence supra-physiological cell iron content.

In any case, even if MINPP1^{-/-} cells do have less free iron, the authors do not further explore how this could offer a mechanistic basis for the PCH condition.

Figures 5A,B,C,D should be removed.

6. The impact of the MINPP KO on calcium signaling is intriguing, but likely not in the oversimplified manner suggested by the authors: increased cation-binding by IP6 (e.g., see the abstract). The latter is an excessive speculation. Indeed, there is a prior publication based on solid physicochemical data that argues IP6 in vivo is mainly chelated with Mg, such that there is negligible binding to other cations, including calcium (9). This point should be noted. One of the authors (AS) has questioned this idea by noting in a recent publication (10) that there is no involvement of Mg in any published crystal complexes of IP6 with a protein, but of course, this does not directly address the issue of IP6/cation status in vivo, and in any case Mg may be stripped upon protein binding. After all, waters of solvation are stripped from ligands that bind proteins (11), so why not cations too?

The authors must consider and explore alternative explanations. Surely it is very likely that a change in cytoplasmic calcium concentration reflects alterations in calcium homeostasis through manipulation of intracellular and/or plasma membrane calcium fluxes. Pursuit of these alternative options could greatly benefit the study by providing a plausible mechanistic basis that is currently lacking.

For example, the authors also report the MINPP1 deletion reduces intracellular calcium pools (i.e., those releasable by caffeine or ionomycin). However, the methods section indicates these experiments were conducted in the presence of extracellular calcium, and so it's possible the

results obtained reflect primary inhibition of calcium entry. Thus, the authors should test this idea by repeating these experiments in the absence of extracellular calcium, and they should separately assess the rate of calcium entry in the MINPP^{-/-} model, which can be readily accomplished by using the manganese quench assay (12).

Inhibition of calcium entry over time would also eventually deplete intracellular stores. So, is there any evidence in the literature that elevated IP5/IP6 may inhibit calcium entry? The authors could also note that complex pathologies result from ER calcium dysregulation (13) that, perhaps, may also underlie the PCH phenotype.

7. On page 13, the authors describe experiments with conditioned media that show secretion of WT MINPP1 from cells; this is an underappreciated phenomenon, since the C-terminal SDEL tetrapeptide in MINPP1 is an effective ER retention signal (1). Nevertheless, Windhorst et al (4) previously demonstrated MINPP1 secretion, and their study should be cited here. The authors do cite the "extracellular" location of *Drosophila mipp1*, but that description should be clarified. The *Drosophila* enzyme is actually attached to the cell surface, and is not free in the extracellular space; the fly enzyme gets to this location by a mechanism different from human MINPP1 secretion.

8. Most of the citations in the text are offset by one digit from the appropriate papers listed in the bibliography.

9. Page 18, sentence beginning on the last line: "[lower free calcium in MINPP^{-/-} cells] was associated with a consistent slight decrease in the level of the calcium-dependent calmodulin protein, illustrating a potential physiological consequence of this change." The authors should delete this excessive speculation that links two events that may not have any cause/effect relationship, unless the authors can quote published literature that show calmodulin expression is normally regulated by free cytoplasmic calcium. As to the apparent 10% decrease in calmodulin expression per se, this needs some statistical clarity: does the accompanying immunoblot in Fig. S5F describe 4 replicates from one experiment (i.e., "technical replicates")? If so, the authors must confirm the data in the bar graph were obtained from 4 replicates of such experiments. Or were the data obtained from different batches of cells, in genuinely separate experiments, that were all run on the blot that is shown (i.e., "biological replicates")?

1. Raykhel I, et al. (2007) A molecular specificity code for the three mammalian KDEL receptors. *J Cell Biol* 179(6):1193-1204.
2. Craxton A, Caffrey JJ, Burkhardt W, Safrany ST, & Shears SB (1997) Cloning and expression of rat hepatic Multiple Inositol Polyphosphate Phosphatase. *bj* 328:75-81.
3. Cho J, et al. (2006) Avian multiple inositol polyphosphate phosphatase is an active phytase that can be engineered to help ameliorate the planet's "phosphate crisis". *J. Biotechnol* 126(2):248-259.
4. Windhorst S, et al. (2013) Tumour cells can employ extracellular Ins(1,2,3,4,5,6)P6 and multiple inositol-polyphosphate phosphatase 1 (MINPP1) dephosphorylation to improve their proliferation. *Biochem. J* 450(1):115-125.
5. Hidaka K, et al. (2003) The importance to chondrocyte differentiation of changes in expression of the multiple inositol polyphosphate phosphatase. *Exp. Cell Res* 290(2):254-264.
6. Otto JC, Kelly P, Chiou ST, & York JD (2007) Alterations in an inositol phosphate code through synergistic activation of a G protein and inositol phosphate kinases. *Proc. Natl. Acad. Sci. U. S. A* 104(40):15653-15658.
7. Chi H, et al. (2000) Targeted deletion of *Minpp1* provides new insight into the activity of multiple inositol polyphosphate phosphatase in vivo. *Mol. Cell. Biol* 20:6496-6507.
8. Anderson GJ & Frazer DM (2017) Current understanding of iron homeostasis. *Am J Clin Nutr* 106(Suppl 6):1559S-1566S.
9. Torres J, et al. (2005) Solution behaviour of myo-inositol hexakisphosphate in the presence of

multivalent cations. Prediction of a neutral pentamagnesium species under cytosolic/nuclear conditions. *Journal of Inorganic Biochemistry* 99:828-840.

10. Wilson MS, Jessen HJ, & Saiardi A (2019) The inositol hexakisphosphate kinases IP6K1 and -2 regulate human cellular phosphate homeostasis, including XPR1-mediated phosphate export. *J Biol Chem*.

11. Shoichet BK (2007) No free energy lunch. *Nat. Biotechnol* 25(10):1109-1110.

12. Pan Z, Choi S, & Luo Y (2018) Mn(2+) Quenching Assay for Store-Operated Calcium Entry. *Methods Mol Biol* 1843:55-62.

13. Trychta KA, Back S, Henderson MJ, & Harvey BK (2018) KDEL Receptors Are Differentially Regulated to Maintain the ER Proteome under Calcium Deficiency. *Cell Rep* 25(7):1829-1840 e1826.

Reviewer #2 (Remarks to the Author):

The manuscript by Ucunu et al describes the identification of MINPP1 mutations in patients with a cerebellar hypoplasia. Three small families with MINPP1 mutations were identified.

The manuscript is well written and has a significant amount of supportive experiments that show that MINPP1 is deficient in cells derived from a patient and the fibroblast derived iPSCs. Despite the fact that MINPP1 deficiency has been shown for at least 1 patient, the evidence that this is the cause for the cerebellar atrophy is not strong.

The segregation of the mutation in the pedigrees is not convincing. Why is the genotype of the healthy sibs in CerID-09 (figure S1) not given? Are MINPP1 sequence variants the only variants that segregate with the phenotype?

The analysis of iPSCs and the derived neurons shows a clear difference between normal and deficient cells, but this phenotype is so strong that it is surprising that the patients only show mild cerebellar atrophy. Surprisingly, the k.o. mice from the current study and Chi et al (MCB, , DOI: 10.1128/MCB.20.17.6496-6507.2000) shows no significant effect on brain development. The reduced cortical thickness in the k.o. mice is not well documented and couldn't this be due a delayed maturation?

The biochemical analysis of the MINPP1 mutants is well presented, but not novel. Minpp1 deficient mice have already been generated in 2000 (Chi et al, MCB). The accumulation of IP6 in MINPP1 deficient cells is convincing, but what does the experiment with 10 day differentiating neurons mean. The authors show that neuronal differentiation is severely affected in the MINPP1 deficient iPSCs.

In summary, no convincing data that the MINPP1 mutations are associated with PCH. An effect on MINPP1 activity and free cation levels is demonstrated.

For the Reviewers:

We thank the Reviewers for their critical but constructive comments. We noticed that Reviewer#1, although critical of the absence of “mechanistic basis”, agrees with the main conclusion of this study that disruptive mutations of the *MINPP1* gene can cause pontocerebellar hypoplasia. We also noticed that Reviewer #2 acknowledges that the biochemical analysis of the *MINPP1* mutants is well presented and that an effect on *MINPP1* activity and free cation levels is demonstrated.

Reviewer #1

The author's central hypothesis is that mutations in MINPP1 lead to alterations in IP5/IP6 levels that perturb neuronal development, leading to pontocerebellar hypoplasia (PCH) in humans. No sound mechanistic basis for the phenotype is advanced. The proposed mechanistic basis – that increased IP6/cation chelation reduces free levels of iron and calcium – arises (in the case of iron) out of poorly designed unphysiological experiments and (in the case of calcium) an oversimplified interpretation of the data, and a failure to consider very plausible but alternative interpretations. In any case, even if there were more cation chelation, it is not obvious how that could lead to the PCH condition.

To some extent, these criticisms could be addressed by further experiments.

Response: The reviewer's introductory assessment is not quite correct. Our work demonstrates that *MINPP1* mutations cause pontocerebellar hypoplasia. This conclusion is now further reinforced by additional analysis as listed below (see response to point 1). The intracellular accumulation of IP₆ is detected as the most consistent imbalance resulting from *MINPP1* mutations. Consequently, this imbalance is proposed as the basis of the mechanism. We agree with the reviewer that cation chelation is not an obvious mechanism for the PCH condition, and that other plausible explanations could exist. Additionally, there are no obvious pathways for PCH with the exception of tRNA splicing (through the TSEN complex). We partially agree that our cation alteration analysis was imperfect, and have performed further experiments to reinforce these data. We have followed and fulfilled the reviewer's conclusive statement “*these criticisms could be addressed by further experiments*”.

Point 1 paragraph 1:

The human genomic evidence of a link between MINPP1 mutations to PCH (Fig. S1) is very preliminary: sequencing was performed for only 10 individuals from 3 families. Also, in family CerID-09, there is an apparent lack of sequencing data for MINPP1 in one PCH individual and 4 unaffected siblings. Such positive and negative control data would allow a more complete picture on the degree to which MINPP1 mutations associate with the PCH phenotype. Presumably such data cannot be obtained, but their absence does weaken the conclusions. To reflect the preliminary nature of these data, the title of Fig. 1 should be modified to indicate that mutations in MINPP1 are associated with the PCH phenotype, not the “cause” of it. Furthermore, in the text (page 5, 4 lines up), the authors state that MINPP1 was just one of an unspecified number of “new candidate genes that were identified.” If there is a stronger association of any of these mutations with the phenotype, they should be described.

Response: In the new version of the manuscript, the genetic data are enforced by substantial additional analysis:

- We now provide three additional families with *MINPP1* bi-allelic variants, identified by two other groups, in patients with PCH. These cases include another homozygous truncating mutation.
- We have included a summary of the genetic results in a new variant table (new Table 1) that includes the frequency and pathogenicity prediction of all the identified variants. These data show that these variants are predicted to be pathogenic, completely absent from control population at the homozygous state, and unique for most of them.

Concerning the incomplete genetic segregation data for family CerID-09, also pointed out by Reviewer #2, we apologize for not providing an explanation previously that is now included in the manuscript. To clarify further, we have removed non-genotyped patient CerID09-01 from the clinical table (now Table 2) and Figure 1. As identified by the experienced reviewer #1, it was impossible to collect additional samples as the family did not wish to participate in the study anymore.

We would like to underline the fact that these *MINPP1* variants were all identified in patients with an extremely specific, never before described, neuroradiological phenotype that includes the combination of PCH and basal ganglia/thalami alterations. For one of the new patients, the *MINPP1* molecular defect was proposed by the neuroradiologist, N. Boddaert, based on the MRI and blinded to the genetic analysis result. These arguments, including the independent identification of this gene defect in similar patients by another group, unquestionably support the fact that *MINPP1* mutations cause PCH.

Regarding the candidate genes, we apologize for the misunderstanding. No other strong candidate genes were identified in the families with *MINPP1* mutations, the candidate genes mentioned were identified in other families. We have now modified the manuscript to clarify this point and we have provided supplementary information concerning the variants identified in the families with *MINPP1* mutations and their analysis (see "Supplementary Note: Detailed Sequencing Variant Filtering and Prioritization").

Point 1 paragraphs 2-5:

Another problem is that among just 4 PCH individuals that provide sequencing data, only 2 of these can currently be considered as providing direct support for the author's hypothesis that changes in MINPP1 catalytic activity underlie PCH: the early truncation mutant for family CerID-30, which surely destroys MINPP activity. To be fair, the PCH condition is very rare, so sample numbers are inevitably low. Nevertheless, the authors have no direct evidence that the 2 other mutations – Tyr53Ala and Glu486Lys – have any impact on MINPP1 activity and IP5/IP6 metabolism in vivo.

However, Tyr53Ala is not too far from the active site and could have a previously unappreciated catalytic impact; or it could indirectly alter activity by affecting protein folding. The latter should at least be studied in silico, perhaps using the Missense3D algorithm of the PHYRE structural prediction server. As for Glu486Lys, it lies in a functionally-distinct, ER retention tetrapeptide (1, 2), which should be noted; if the effect of that mutation is to enhance MINPP1 secretion, it could reduce the amount of cellular enzyme that has access to substrates, also raising IP5/IP6 – i.e., phenocopying the MINPP1 knockout. This issue is so central to the author's hypothesis that they must directly assess the catalytic activities of recombinant versions of the Tyr53Ala and Glu486Lys mutants and/or the impact of their over-expression on IP5/IP6 levels in intact cells. The apparent absence of an effect of these two mutations upon cell growth (Fig. 2E) does not provide any direct evidence as to the state of the catalytic activity.

The subject of MINPP1 secretion also needs expanding slightly, because Fig. S2D appears to show that overexpressed Tyr53Ala and Glu486Lys migrate as a doublet following Western analysis, while WT enzyme appears as a single band. This, too, deserves comment. The doublet likely reflects glycosylation as MINPP1 progresses through the secretory pathway (3, 4), consistent with the possibility (see above) that the Glu486Lys mutation is a gain-of-secretion mutant. The authors should therefore study this possibility by assessing levels and catalytic activities of the two mutants in conditioned medium – are they different from secreted WT enzyme?

If on the other hand the authors were to establish one or both of the Tyr53Ala and Glu486Lys mutations do not alter cellular IP5/IP6 levels, they may wish to conclude that they have a false-positive association with the PCH condition.

Response: The reviewer correctly requested evidence on the loss of catalytic activity of the *MINPP1* mutants identified. As suggested, we now provide *in silico* analysis using Missense3D (Supp. Fig. 1B). The software predicts that all the variants involving amino acids included in the model (i.e. Tyr53, Phe228, Arg410 but not Glu486 that is not present in the *D. castellii* phytase crystallized structure) causes structural damages to the *MINPP1* protein.

Using *in vivo* over-expression after 3H-inositol radiolabeling with Sax-HPLC analysis (see new Sup. Figure 4 C-D), we found a completely abrogated enzyme activity for the Tyr53Ala mutation. However, Glu486Lys was catalytically active in this over-expression setup. As intelligently highlighted by the reviewer, this variant could instead be "a gain-of-secretion mutant": as the Glu486Lys variant is located in the ER retention signal peptide, the mutant proteins may be secreted or otherwise mislocalized into some compartment without access to its IP₆ substrate. Perhaps *MINPP1* activity/IP₆ degradation specifically within the ER lumen is required for healthy cells. The development of a functional, tagged version of Glu486Lys *MINPP1*, without altered ER targeting and ER retention, is of our interest to study the stability, localization and secretion of this variant protein. However, we believe that these analyses are out of the scope of the current study.

Nevertheless, pathogenicity of the Glu486Lys variant is extremely likely, based on 1) the very similar clinical phenotype of the CerID09 family versus the others, 2) the genetic investigation of the CerID09 family, 3) the clear inability of this variant to rescue the proliferation phenotype in HEK293 cells, 4) the potential reduced stability/mislocalization associated with this variant.

Regarding post-translational modification of the over-expressed MINPP1, we have performed additional experiments to investigate this as requested. We observed a doublet by Western blot for all the over-expressed forms, including the WT, but never for the endogenous protein suggesting an artefact related to the transfected MINPP1. We also performed PNGase treatment to study N-glycosylation (1). Lower MW forms, representing removal of glycan chain(s), appeared similarly for WT and variant MINPP1, suggesting that no large differences in modifications were present (Sup. Fig. 2E). However, the new blots, with equal protein loading, clearly indicate lower levels of over-expressed mutant proteins than the WT (new Sup Fig 2D). The mutations could therefore impact MINPP1 stability.

Point 1 paragraph 6:

The authors should also report the results of screening appropriate databases for the incidence of each of the three mutations they have described.

Response: We thank the reviewer for this important suggestion. We have added a new table (new Table 1), indicating the incidence of the variants observed in this study.

Point 2:

The partial rescue of the MINPP1--induced proliferation defect by overexpression of wild-type MINPP1 is of potential interest, but the authors must also perform the positive control, i.e., determine if MINPP1 over-expression at least partially restores cellular IP5/IP6 levels. This experiment is made even more vital by a prior observation that WT MINPP1 over-expression in ER does not alter IP5/IP6 levels (5). Obtaining these metabolic data for wild-type enzyme is also a necessary control for studying the effects of the Tyr53Ala and Glu486Lys mutations upon IP5/IP6 levels (see point 1).

Response: We thank the reviewer for the correct comment. We now provide clear evidence of rescue of IP₆/IP₂ levels in MINPP1 KO cells (Sup. Fig. 4C,D). The experiment mentioned, where no effect was seen, was performed in WT cells.

Point 3:

page 18: "we identified a mild but significant ~10% decrease in the brain weight associated with a reduced cortical thickness in [minpp ko] mice at P21. This observation suggests the presence of an evolutionarily conserved requirement for MINPP1 activity in mammalian brain development, despite apparent differences in the severity of the phenotype." This is rather an exaggeration; a slightly smaller brain size and a thinner cortex is not a recapitulation of a less severe PCH phenotype. Delete the phrase "despite apparent differences in the severity of the phenotype."

Some might argue that the inability of the minpp-null mice to recapitulate the human PCH condition is of detriment to the study, particularly since (see point 1), the human genetic data are themselves very preliminary. However, the central concept that MINPP1 deletion compromises neuronal differentiation is supported by data in Figs 3,4 that show patient-derived and MINPP-null iPSCs exhibit elevated IP6 levels and impaired neuronal differentiation. And in any case, human brains are different from mouse brains. While mice brain development may be able to (largely) compensate for minpp loss, and humans cannot (a point that could be emphasized), isn't that in itself a significant and interesting observation? That being said, these data from iPSCs only validate an effect of the MINPP1-null condition, not the Tyr53Ala and Glu486Lys mutations; this point should be noted.

Response: We have deleted the requested phrase in the text and added clarification. Other current evidences related to PCH modelling show that the severity of the human phenotype is not recapitulated in mouse. MINPP1 mutants can be another example. We agree with the reviewer and now make more emphasis on the potential human-specific sensitivity in the discussion. The complete loss of function of MINPP1 in PCH is now directly supported by another family with null mutation (PCH-2712) and with functional assessment of Tyr53Ala activity.

Point 4:

The authors construct a hypothesis concerning cation-IP6 binding that ignores complexities and uncertainties concerning both the disposition of cellular IP6 and its access to MINPP1. The authors need to be more transparent about published data that challenge the simplicity of their ideas.

(a) For example, page 3, 4-5 lines up. “[IP6 has a] cytosolic concentration of 50-100 μM ”. This misquotes a statement from citation 5; the latter actually describes the range as being 15-100 μM , and goes on to state that this range represents estimates of total cellular IP6 concentration, and, furthermore, “much of this InsP6 may not be freely soluble.” More recent work has established that IP6 is a structural cofactor for certain proteins, further reducing its free levels. The author’s text should be clarified to avoid it being misinterpreted as meaning free cytosolic IP6 exists at up to 100 μM . The actual free IP6 concentration is unknown, but likely far less than 100 μM .

(b) Page 4, 7-9 lines up. “The dynamic regulation of the endogenous pool of IP6 is not fully understood.” [my emphasis]. This implication that there is just one cellular pool of IP6 is not correct. IP6 that acts as a structural cofactor is surely not in the same pool as cytoplasmic IP6. Also, Otto et al (6) have published good evidence IP6 exists in different metabolic pools. Does each pool have equal access to MINPP1?

(c) On page 8 the authors correctly describe cellular MINPP1 as being localized in the ER lumen. It should also be noted that it’s unclear how MINPP1 accesses its substrates. That background better justifies why the authors appropriately highlight their observation that there are higher cellular levels of MINPP1 substrates when the enzyme is knocked-out (page 14, last sentence). Nevertheless, they should note here that others have previously described this very phenomenon (7).

Response: The field related to the MINPP1 enzyme includes major discrepancies and uncertainties (e.g. cellular localization of the enzyme and its substrates, as the reviewer points out). Some were not developed in the manuscript, not because of a lack of transparency but only to improve the manuscript readability, when we don’t address directly these issues. We now stress more the unsolved question related to the access of MINPP1 to its substrate, referencing the appropriate literature, although we don’t address it directly.

We apologize for the mistake and have corrected the sentence as saying IP₆ has a cellular concentration of 15-100 μM . We would like to clarify that we never mentioned “free IP₆ levels” in our text. We agree with the reviewer that IP₆ could be present in association with other proteins, and the free levels are unknown.

Regarding “the endogenous pool of IP6”: as mentioned in the title, and differently to what has been observed with a *Drosophila* homolog (2) or suspected in some cancer cells (3), we want to emphasize the role of the mammalian MINPP1 on the intracellular IP₆ levels (endogenous) vs extracellular (artificially added/exogenous). This sentence was meant to highlight this point. We have now re-phrased the sentence, and also mentioned the likely presence of different intracellular pools of IP₆.

We have now also discussed the discrepancy between the localization of MINPP1 and its substrates. As to whether others have described “the very phenomenon” of increased MINPP1 substrates in KO cells, it is true that Chi et al. previously investigated endogenous IP₅ and IP₆, and reported a small 30% increase of IP₆ in *Minpp1* KO embryonic fibroblasts. That paper, which we cite, was limited to analysis of only IP₅ and IP₆. We are reporting the investigation of more potential substrates and products (7 different IPs), with higher sensitivity, in a different species (human), and in multiple cell types. We therefore report a more complete signature of IPs imbalances, with robust ~3-fold and ~2-fold IP₆ increases in HEK293 and differentiating neurons respectively, in the absence of MINPP1. Importantly, by analyzing the whole inositol phosphates profile, we discovered the never reported decrease in IP₂ level in *MINPP1* KO cells. This is a highly unexpected result, with far-reaching implications for understanding IPs metabolism. This highlights the thorough nature of our study and novelty of our discoveries. Given the differences in the species, the method used, the metabolites analyzed, the original discovery and their implications, we reject the idea that our results have been shown before.

Point 5 paragraphs 1-2:

In Fig. 2A, the authors assay total cellular iron in WT and MINPP1^{-/-} cells. Under physiologically-relevant, low-iron conditions, there is no impact of the MINPP1^{-/-} genotype upon iron levels (it’s hard to tell from the Y-axis scaling, but the knockout cells may actually have less iron). But there is a qualitatively different outcome when the authors incubate the cells for 48 hours with unphysiologically high extracellular iron concentration (100 μM): then, the knockout cells accumulate more iron. The reliance of this experimental result on purely non-physiological conditions (that greatly perturbs iron homeostasis (8)) renders its conclusions unreliable.

See Fig. 5C,D. There is a fundamental experimental flaw in the efforts to assay free cellular Fe²⁺ and Fe³⁺; it is not possible to obtain the data in intact cells, so the authors pursue this goal with cell

extracts. Imagine the furor that would result if a journal published a paper that used the same approach to record free cytoplasmic calcium!

Response: To study cellular iron homeostasis we used established and widely used experimental conditions (4-7). In Fig. 5A, the concentration of FAC that is used is the same as what is used in a considerable number of studies to artificially recapitulate iron accumulation defects, as in many models the phenotype is not visible in the absence of an excess of iron (8-10). Similarly, the techniques used to quantify iron in whole cell extracts here are widely used and accepted (11-17). We apologize for a mistake in the subtitle p.17 where “cytosolic free iron” was mistakenly written instead of “free iron”. This mistake has been corrected. In Fig 5B, for clarity the legend of the y-axis has been changed to “Free Fe₂₊ + Fe₃₊”.

Point 5 paragraphs 3-5:

Just as is the case for calcium, iron is present in different cellular pools. In intact cells, iron is distributed between cytoplasm, mitochondria, lysosomes and endosomes; much cytoplasmic iron is stored within ferritin complexes (8). Very little iron is actually ‘free’. Moreover (point 4), IP6 appears to be compartmentalized too. After cell lysis, this compartmentalization is largely destroyed. Under such in vitro conditions, when IP6 and iron are now free from compartmentalization, the known tight chelation of iron by IP6 binding is expected to materialize. Thus, the cell lysates that contain the most IP6 (i.e., those prepared from MINPP1^{-/-} cells) will also exhibit the least free iron. That is, the experiments described in Fig 2C,D are “doomed to succeed.” The resulting in vitro data are not evidence of IP6 reducing free cytoplasmic iron levels in intact cells as a mechanism of action relevant to neurological disorders. Note also (see above), such data are obtained in the context of cell preincubation in an unphysiologically high extracellular iron concentration and hence supra-physiological cell iron content.

In any case, even if MINPP1^{-/-} cells do have less free iron, the authors do not further explore how this could offer a mechanistic basis for the PCH condition.

Figures 5A,B,C,D should be removed.

Response: We partially disagree with the reviewer (we assume that the “Fig 2C,D” mentioned refers to Fig.5C,D). At pH below 3.5 IP₆ is soluble and can't complex with Fe₃₊ (18-19). For these experiments the protocol (different from Fig.5A) uses a lysis buffer at pH 4.6, which is likely to maintain IP₆ complexed with its intracellular bivalent cation partner/s primarily magnesium (20). Thus, our experiments are not “doomed to succeed”. Additionally, the predicted “post-lysis chelation” cannot explain the increase of total cellular iron observed in Fig.5A

These experiments should be interpreted as in vitro evidence of the potential impact of MINPP1 absence (and IP₆ accumulation) on cellular cations, but we agree should not be directly interpreted in regards to the disease pathology. Consequently, we have not removed these data, but have followed the reviewer's advice to improve their interpretation in the Discussion.

Point 6:

The impact of the MINPP KO on calcium signaling is intriguing, but likely not in the oversimplified manner suggested by the authors: increased cation-binding by IP6 (e.g., see the abstract). The latter is an excessive speculation. Indeed, there is a prior publication based on solid physicochemical data that argues IP6 in vivo is mainly chelated with Mg, such that there is negligible binding to other cations, including calcium (9). This point should be noted. One of the authors (AS) has questioned this idea by noting in a recent publication (10) that there is no involvement of Mg in any published crystal complexes of IP6 with a protein, but of course, this does not directly address the issue of IP6/cation status in vivo, and in any case Mg may be stripped upon protein binding. After all, waters of solvation are stripped from ligands that bind proteins (11), so why not cations too?

The authors must consider and explore alternative explanations. Surely it is very likely that a change in cytoplasmic calcium concentration reflects alterations in calcium homeostasis through manipulation of intracellular and/or plasma membrane calcium fluxes. Pursuit of these alternative options could greatly benefit the study by providing a plausible mechanistic basis that is currently lacking.

For example, the authors also report the MINPP1 deletion reduces intracellular calcium pools (i.e., those releasable by caffeine or ionomycin). However, the methods section indicates these experiments were conducted in the presence of extracellular calcium, and so it's possible the results obtained reflect primary inhibition of calcium entry. Thus, the authors should test this idea by repeating these experiments in the absence of extracellular calcium, and they should separately assess the rate

of calcium entry in the *MINPP*^{-/-} model, which can be readily accomplished by using the manganese quench assay (12).

Inhibition of calcium entry over time would also eventually deplete intracellular stores. So, is there any evidence in the literature that elevated IP₅/IP₆ may inhibit calcium entry? The authors could also note that complex pathologies result from ER calcium dysregulation (12) that, perhaps, may also underlie the PCH phenotype.

Response: We are pleased that the reviewer is interested by our data on the impact of *MINPP1* on calcium homeostasis. As highlighted by the reviewer, how much calcium binds to IP₆ is a question still open, especially as the study mentioned (20) is predictive. Additionally, it cannot be predicted if in the absence of *MINPP1*, the accumulated IP₆ will behave the same way as the “normal” ~50 μM IP₆ present in WT cells, or accumulate in a different pool/location.

Nevertheless, as suggested we have performed additional calcium assays. We now present data using caffeine or ionomycin both in the presence (Fig 5G,H) and absence of extracellular calcium (21; new Supplementary Fig 5F,G), in WT HEK293, *MINPP1* KO, or new *MINPP1* KO cells stably expressing *MINPP1* WT. Overexpression of *MINPP1* partially rescued the phenotype, and similar results were seen when extracellular calcium was present or absent, indicating that the defect that we observe is not related to a primary inhibition of calcium entry. As there is no direct experimental or previously published evidence that IP₆ could block calcium entry, we favor other hypotheses that implies an impact of *MINPP1* on ER-stored calcium.

Point 7. On page 13, the authors describe experiments with conditioned media that show secretion of WT *MINPP1* from cells; this is an underappreciated phenomenon, since the C-terminal SDEL tetrapeptide in *MINPP1* is an effective ER retention signal (1). Nevertheless, Windhorst et al (4) previously demonstrated *MINPP1* secretion, and their study should be cited here.

The authors do cite the “extracellular” location of *drosophila mipp1*, but that description should be clarified. The *drosophila* enzyme is actually attached to the cell surface, and is not free in the extracellular space; the fly enzyme gets to this location by a mechanism different from human *MINPP1* secretion.

Response: We thank the reviewer for this comment. The Windhorst et al. study, which we do cite elsewhere, is unfortunately unclear regarding *MINPP1* secretion. They lack a clear negative control i.e. *MINPP1* KO cells), and rely on knock-down combined with the addition of exogenous (non-physiological) IP₆. Consequently, it is difficult to interpret their inconsistent WB data (Fig.6C) and IP_s quantification (Fig.7B); we therefore prefer not to cite this work for this specific point (3). We have, however, followed the reviewer’s helpful suggestion to clarify the text and have replaced “extracellularly” with “outside the cell”.

Point 8:

Most of the citations in the text are offset by one digit from the appropriate papers listed in the bibliography.

Response: We thank the reviewer for noticing. The corrected version is now provided.

Point 9:

Page 18, sentence beginning on the last line: “[lower free calcium in *MINPP*^{-/-} cells] was associated with a consistent slight decrease in the level of the calcium-dependent calmodulin protein, illustrating a potential physiological consequence of this change.” The authors should delete this excessive speculation that links two events that may not have any cause/effect relationship, unless the authors can quote published literature that show calmodulin expression is normally regulated by free cytoplasmic calcium. As to the apparent 10% decrease in calmodulin expression per se, this needs some statistical clarity: does the accompanying immunoblot in Fig. S5F describes 4 replicates from one experiment (i.e., “technical replicates”)? If so, the authors must confirm the data in the bar graph were obtained from 4 replicates of such experiments. Or were the data obtained from different batches of cells, in genuinely separate experiments, that were all run on the blot that is shown (i.e., “biological replicates”)?

Response: For the data obtained in Fig.S5F, N referred to biological replicates (here different mice). Although the statistical significance was clear, the reviewer is correct to point out that the interpretation of a link is difficult. To avoid confusion, we have removed panel F from Sup. Fig5.

Reviewer #2

The manuscript by Ucunu et al describes the identification of MINPP1 mutations in patients with a cerebellar hypoplasia. Three small families with MINPP1 mutations were identified.

The manuscript is well written and has a significant amount of supportive experiments that show that MINPP1 is deficient in cells derived from a patient and the fibroblast derived iPSCs. Despite the fact that MINPP1 deficiency has been shown for at least 1 patient, the evidence that this is the cause for the cerebellar atrophy is not strong.

The segregation of the mutation in the pedigrees is not convincing. Why is the genotype of the healthy sibs in CerID-09 (figure S1) not given? Are MINPP1 sequence variants the only variants that segregate with the phenotype?

Response: As discussed in the answer to Reviewer 1 point 1 paragraph 1, additional samples from family CerID-09 could not be collected as the family decided to stop collaborating, which is not unusual in such a devastating medical context. The human genetics have been strengthened with new families, new mutations (new Fig.1, Table 1, Table 2) and new data (new Fig.5; sup. Fig. 1B; sup. Fig. 2D, sup. Fig.4 C,D; sup. Fig.5 F, G) that support the previous findings. MINPP1 sequence variants are indeed the only solid candidate variants segregating with the phenotype. We have now provided additional information about variants filtering and prioritization (see new section Supplementary Note).

The analysis of iPSCs and the derived neurons shows a clear difference between normal and deficient cells, but this phenotype is so strong that it is surprising that the patients only show mild cerebellar atrophy.

Response: We agree that the phenotype shown is very striking. However, it is difficult to compare the in vivo developmental outcome of a metabolic defect with in vitro cultured cells. It cannot be excluded that neurons out of their physiological environment show an exacerbated phenotype. Interestingly, cell-specific differences in the expression of enzymes or transporters involved in the inositol metabolic pathway have previously been reported in the nervous system (22-26).

Surprisingly, the k.o. mice from the current study and Chi et al (MCB, DOI: 10.1128/MCB.20.17.6496-6507.2000) shows no significant effect on brain development. The reduced cortical thickness in the k.o. mice is not well documented and couldn't this be due a delayed maturation?

Response: As described, we identified a significant difference in the brain weight, in addition to reduced cortical thickness. Interestingly, a comparable difference was observed in the mouse model for another PCH gene, *CLP1* (27). We have now added more data on brain weight differences at later time points (new sup. Fig. 5D): the differences still persist with similar difference in the brain weight. At 11 month old, the decrease of the average brain weight of the homozygous mutant mice was 12.5%, which is very similar to that at p21 (12.7%), suggesting an early defect occurring during development versus a delayed maturation.

The biochemical analysis of the MINPP1 mutants is well presented, but not novel. Minpp1 deficient mice have already been generated in 2000 (Chi et al, MCB).

Response: As discussed above (Reviewer 1 point 4), the differences between our study and the work mentioned are very important, and the correlations between the cellular, mouse and clinical phenotypes with the biochemical data here provide a completely novel context. Furthermore, we would like to reiterate that the decrease in IP₂ observed is completely unexpected and novel. This finding could lead to the redesign of our understanding of inositol phosphates metabolic fluxes.

The accumulation of IP6 in MINPP1 deficient cells is convincing, but what does the experiment with 10 day differentiating neurons mean. The authors show that neuronal differentiation is severely affected in the MINPP1 deficient iPSCs.

Response: The day 10 differentiation point was chosen to select a stage just after neural induction (i.e. at day 7 when we start the metabolic labelling) and before the observation of differences in the TUJ1/PAX6 cell populations. We have now added this explanation to the method part.

In summary, no convincing data that the MINPP1 mutations are associated with PCH. An effect on MINPP1 activity and free cation levels is demonstrated.

Response: We thank the reviewer for recognizing the effect on MINPP1 on cation physiology. We have now have provided further data and solid genetic evidence (as described above in the first response to reviewer 2, and reviewer 1 point 1) that unquestionably associate MINPP1 mutations with PCH.

References:

1. Medina-Cano, D. *et al.* High N-glycan multiplicity is critical for neuronal adhesion and sensitizes the developing cerebellum to N-glycosylation defect. *Elife* 7(2018).
2. Cheng YL, Andrew DJ. Extracellular Mipp1 Activity Confers Migratory Advantage to Epithelial Cells during Collective Migration. *Cell Rep.* 2015 Dec. 15;13(10):2174-88.
3. Windhorst, S. *et al.* Tumour cells can employ extracellular Ins(1,2,3,4,5,6)P(6) and multiple inositol-polyphosphate phosphatase 1 (MINPP1) dephosphorylation to improve their proliferation. *Biochem J* **450**, 115-25 (2013).
4. Jiang Y, Li C, Wu Q, An P, Huang L, Wang J, Chen C, Chen X, Zhang F, Ma L, Liu S, He H, Xie S, Sun Y, Liu H, Zhan Y, Tao Y, Liu Z, Sun X, Hu Y, Wang Q, Ye D, Zhang J, Zou S, Wang Y, Wei G, Liu Y, Shi Y, Eugene Chin Y, Hao Y, Wang F, Zhang X. Iron-dependent histone 3 lysine 9 demethylation controls B cell proliferation and humoral immune responses. *Nat Commun.* 2019 Jul 3;10(1):2935.
5. Ren J, Ding L, Xu Q, Shi G, Li X, Li X, Ji J, Zhang D, Wang Y, Wang T, Hou Y. LF-MF inhibits iron metabolism and suppresses lung cancer through activation of P53-miR-34a-E2F1/E2F3 pathway. *Sci Rep.* 2017 Apr 7;7(1):749.
6. Zhang DL, Hughes RM, Ollivierre-Wilson H, Ghosh MC, Rouault TA. A ferroportin transcript that lacks an iron-responsive element enables duodenal and erythroid precursor cells to evade translational repression. *Cell Metab.* 2009 May;9(5):461-73.
7. Meyron-Holtz EG, Ghosh MC, Rouault TA. Mammalian tissue oxygen levels modulate iron-regulatory protein activities in vivo. *Science.* 2004 Dec 17;306(5704):2087-90.
8. Drecourt A, Babbior J, Dussiot M, Petit F, Goudin N, Garfa-Traoré M, Habarou F, Bole-Feysot C, Nitschké P, Ottolenghi C, Metodiev MD, Serre V, Desguerre I, Boddaert N, Hermine O, Munnich A, Rötig A. Impaired Transferrin Receptor Palmitoylation and Recycling in Neurodegeneration with Brain Iron Accumulation. *Am J Hum Genet.* 2018 Feb 1;102(2):266-277.
9. Das SK, Wang W, Zhabyeyev P, Basu R, McLean B, Fan D, Parajuli N, DesAulniers J, Patel VB, Hajjar RJ, Dyck JR, Kassiri Z, Oudit GY. Iron-overload injury and cardiomyopathy in acquired and genetic models is attenuated by resveratrol therapy. *Sci Rep.* 2015 Dec 7;5:18132.
10. Zheng Q, Zhao Y, Guo J, *et al.* Iron overload promotes mitochondrial fragmentation in mesenchymal stromal cells from myelodysplastic syndrome patients through activation of the AMPK/MFF/Drp1 pathway. *Cell Death Dis.* 2018;9(5):515.
11. Yoshida M *et al.* Involvement of cigarette smoke-induced epithelial cell ferroptosis in COPD pathogenesis. *Nat Commun* 10:3145(2017)
12. Zheng Y *et al.* Expression of β -globin by cancer cells promotes cell survival during blood-borne dissemination. *Nat Commun* 8:14344 (2017).
13. Zhang Y, Schmid B, Nikolaisen NK, *et al.* Patient iPSC-Derived Neurons for Disease Modeling of Frontotemporal Dementia with Mutation in CHMP2B. *Stem Cell Reports.* 2017;8(3):648–658.
14. Alvarez LA, Kovačič L, Rodríguez J, Gosemann JH, Kubica M, Pircalabioru GG, Friedmacher F, Cean A, Ghișe A, Sărăndan MB, Puri P, Daff S, Plettner E, von Kriegsheim A, Bourke B, Knaus UG. NADPH oxidase-derived H₂O₂ subverts pathogen signaling by oxidative phosphotyrosine conversion to PB-DOPA. *Proc Natl Acad Sci U S A.* 2016 Sep 13;113(37):10406-11
15. Li H, Zhang C, Shen H, Shen Z, Wu L, Mo F, Li M. Physiological stress-induced corticosterone increases heme uptake via KLF4-HCP1 signaling pathway in hippocampus neurons. *Sci Rep.* 2017 Jul 18;7(1):5745
16. Healy S, McMahon J, Owens P, FitzGerald U. Significant glial alterations in response to iron loading in a novel organotypic hippocampal slice culture model. *Sci Rep.* 2016 Nov 3;6:36410. doi: 10.1038/srep36410. PubMed PMID: 27808258; PubMed Central PMCID: PMC5093415.
17. Nairz M, Ferring-Appel D, Casarrubea D, Sonnweber T, Viatte L, Schroll A, Haschka D, Fang FC, Hentze MW, Weiss G, Galy B. Iron Regulatory Proteins Mediate Host Resistance to Salmonella Infection. *Cell Host Microbe.* 2015 Aug 12;18(2):254-61.
18. Shafey, T. M., M. W. McDonald, and J. G. Dingle. 1991. Effects of dietary Ca and available phosphorus on digesta pH and on the availability of Ca, iron, magnesium and zinc from the intestinal contents of meat chicken. *Br. Poult. Sci.* 32:185–194.

19. Bretti C, Cigala RM, Lando G, Milea D, Sammartano S. Sequestering ability of phytate toward biologically and environmentally relevant trivalent metal cations. *J Agric Food Chem.* 2012;60(33):8075–8082. doi:10.1021/jf302007v
20. Torres J, et al. (2005) Solution behaviour of myo-inositol hexakisphosphate in the presence of multivalent cations. Prediction of a neutral pentamagnesium species under cytosolic/nuclear conditions. *Journal of Inorganic Biochemistry* 99:828-840.
21. Tong J, Du GG, Chen SR, MacLennan DH. HEK-293 cells possess a carbachol- and thapsigargin-sensitive intracellular Ca²⁺ store that is responsive to stop-flow medium changes and insensitive to caffeine and ryanodine. *Biochem J.* 1999 Oct 1;343 Pt 1:39-44.
22. Di Daniel, E., Cheng, L., Maycox, P. R. & Mudge, A. W. The common inositol- reversible effect of mood stabilizers on neurons does not involve GSK3 inhibition, myo-inositol-1-phosphate synthase or the sodium-dependent myo- inositol transporters. *Mol. Cell. Neurosci.* 32, 27–36 (2006).
23. Wong, Y. H., Kalmbach, S. J., Hartman, B. K. & Sherman, W. R. Immunohisto- chemical staining and enzyme activity measurements show myo-inositol-1- phosphate synthase to be localized in the vasculature of brain. *J. Neurochem.* 48, 1434–1442 (1987).
24. Uldry, M. et al. Regulated exocytosis of an H⁺/myo-inositol symporter at synapses and growth cones. *EMBO J.* 23, 531–540 (2004)
25. Maallem, S., Berod, A., Mutin, M., Kwon, H. M. & Tappaz, M. L. Large discrepancies in cellular distribution of the tonicity-induced expression of osmoprotective genes and their regulatory transcription factor TonEBP in rat brain. *Neuroscience* 142, 355–368 (2006)
26. Guo, W. et al. Developmental regulation of Na⁺/myo-inositol cotransporter gene expression. *Brain. Res. Mol. Brain. Res.* 51, 91–96 (1997).
27. Karaca E, Weitzer S, Pehlivan D, Shiraishi H, Gogakos T. Human CLP1 mutations alter tRNA biogenesis affecting both peripheral and central nervous system function. *Cell.* 2014 Apr 24;157(3):636-50.

REVIEWER COMMENTS

Reviewer #1 (Remarks to the Author):

It is my opinion that the authors have constructively addressed the major concerns of both of the reviewers in this revised manuscript; I recommend acceptance for publication, although I have two minor issues with citations, one of which likely will confuse the reader on a small but significant scientific point. I know (from experience!) it is irritating for an author to receive requests for a second revision, but as I indicate below, the reader will be confused and inadequately informed if these two points are not addressed.

1. Page 15, lines 12-13. The authors comment on the near-WT activity of the E486 missense variant expressed in intact cells:

"The E486K variant did not affect enzyme activity. . . in line with the previously demonstrated preserved enzyme activity in the absence of the ER retention peptide (30)."

The latter is not an appropriate citation. The quotation above strongly implies that there is no effect upon MINPP activity in intact cells upon solely removing the C-terminal ER retention sequence (which, incidentally, is a 'sequence', or a 'motif' or 'signal', not a peptide). If removal of ER retention sequence was all that was done in ref 30, the MINPP construct would still have been delivered to the ER. Instead, the MINPP construct described in ref 30 had been further engineered for removal of the N-terminal ER-targeting sequence. Consequently, the enzyme was shown to be mis-targeted to the cytosol.

That is, ref 30 describes an enzyme mislocalized to the cytoplasm (with increased access to substrate), whereas the author's E486K construct would be delivered to the ER (where access to substrate is uncertain).

So, the author's data are certainly NOT "in line with previously demonstrated" observations in ref 30.

However, there is another publication in the literature, albeit with avian rather than human MINPP, that does indeed demonstrate there is no effect upon MINPP activity in intact cells upon solely removing the C-terminal ER retention sequence: PMID 16759730. The latter paper should be cited at this point in the ms, instead of ref 30.

2. Page 5, 9-11 lines up. The authors emphasize novelty by claiming that "no Mendelian disorder has been shown to be caused by an imbalance in the cytosolic inositol polyphosphate pathway, with the exception of a single variant in a gene involved in the conversion of the pyrophosphates. . ." (my emphasis).

In fact, a second variant of the same gene has recently been described that also is associated with a separate Mendelian disorder of inositol phosphate turnover imbalance. See PMID 31852976. Because that latter study adds a little extra perspective to the novelty claim, it should also be cited.

Reviewer #2 (Remarks to the Author):

The points raised by the reviewer have been addressed. The inclusion of additional data has strengthened the manuscript

Reviewer #3 (Remarks to the Author):

In this manuscript by Ucuncu and Rajamani et al., the authors describe mutations in the multiple inositol polyphosphate phosphatase 1 gene (MINPP1) as causative of Pontocerebellar Hypoplasia (PCH) in patients from three different families. In order to investigate how mutations in MINPP1 lead to this early-onset neurodegenerative syndrome, the authors investigate the accumulation of highly phosphorylated inositols (e.g. IP6) in HEK293 cells, patient fibroblasts, patient-derived iPSCs, genome-edited MINPP1^{-/-} iPSCs as well as iPSC-differentiating neurons. The manuscript is well written and the data is presented in a clear fashion. There is substantial data pointing to dysregulation in inositol phosphate metabolism associated with mutations in MINPP1 in HEK293, iPSCs and immature neurons. However, it is important to highlight that, while the overall story is intriguing and the subject is novel and relevant, there are concerns that should be addressed regarding the stem cell model used in this work.

Major Issues:

1- To understand how mutations in MINPP1 could promote neurological dysfunctions observed in PCH patients, the authors attempted to differentiate iPSCs into cortical neurons. However, the differentiation was unsuccessful using a classic dual-SMAD inhibition protocol due to a lack of neural progenitor cells for the lines with MINPP1 mutation. The authors stated that this event "suggests a critical role for MINPP1 during neuroectodermal induction". This interpretation, thus, suggests that the mutations in MINPP1 studied in this experiment would model a possible neurodevelopmental disease as opposed to the neurodegenerative syndrome which is this type of PCH. This discrepancy between the data interpretation and the disease pathogenesis raises questions about how adequate is this model for this study and should be explored in the discussion.

2- The authors then use noggin as a single SMAD inhibitor and could only then successfully generate neurons. The comparison of mutant vs control lines showed a significant decrease in the percentage of TUJ1+ cells. To the same point of issue 1, given that the model is of a neurodegenerative disorder, it is important to understand if the progenitors (PAX6+) cells are not fully differentiating into neurons or if the cells differentiate until day 10+ and then degenerate. A cell death assay with time points post day 10 could help elucidate this issue.

3- The manuscript also lacks critical characterization of the iPSC differentiated neuronal cultures. Characterization of neural progenitor cells should be performed and should include Nestin in addition to PAX6. Fully differentiated neurons should also be characterized with at least MAP2 in addition to TUJ1. Most importantly, the experiments of IP6 accumulation was performed in d10 immature neurons, however, there is no representative image, in the main figures or supplementary, that show the neural cultures on that day. Additionally, it is only in the methods that the authors provide an explanation as to why d10 was chosen, this information should be present in the result section.

Minor Issues:

1- The authors were careful not to overstate the neuronal cultures due to its immaturity, calling them "differentiating neurons" throughout the manuscript with the exception of the title on page 14: "Inositol polyphosphate metabolism is altered in HEK293, iPSCs and induced neurons mutated for MINPP1". "Induced neurons" should be substituted for immature neurons or differentiating neurons.

2- It is unclear why patient CERID-30-2 was chosen to be reprogrammed and this information should be present.

3- Moreover, the patient CERID-30-2 information describes MINPP1 and MTFMT as “genes with most likely pathogenic variants” however, MTFMT was not explored in the discussion.

Response to Reviewers

Manuscript NCOMMS-19-20655A-Z

“MINPP1 prevents intracellular accumulation of the cation chelator inositol hexakisphosphate and is mutated in Pontocerebellar Hypoplasia”

We thank the Reviewers for their appreciation of the novelty and relevance of the work, for agreeing that we have addressed the major concerns from the previous version, and for providing constructive comments. Following the Reviewers and Editor suggestions, we have included new experiments and analysis, which we feel have greatly improved the manuscript.

We have provided additional quality controls for each of the four iPSC lines and studied cell survival in differentiating neurons.

Specifically, we have provided assessment of pluripotency with embryoid body characterization (Supplementary Fig. 3E) and analysis of additional pluripotency markers SSEA4 and TRA-1-81 (Supplementary Fig. 3B), proliferation (Supplementary Fig. 3D) and apoptosis assays (Fig.3D). We have included additional information about cloning and reprogramming efficiency (CerID-30-2 and Ctrl-I004 lines) and the origin of the Ctrl-D10 line (Material and Methods). All of the results indicate an appropriate quality and integrity of the four lines used for this study.

Our analysis of cell survival in differentiating neurons found a very significant increase in apoptosis in differentiating neurons mutated for *MINPP1*, which was not detected in the corresponding iPSCs (Fig.3 C, D). The combination of the inefficient neuronal differentiation with this cell survival defect adds an additional insight to understanding the pathogenesis of this poorly understood very-early onset neurodegenerative disorder (i.e. pontocerebellar hypoplasia).

We have also corrected a small mistake in the pedigree TR-PCH-01 (Supplementary Fig.1 A) and provided additional data reinforcing the significance of the results of the mouse histological study (Supplementary Fig.5 D). To acknowledge the contribution of a new author, Celine Banal, for the embryoid body characterization, we have added her name in the authors list.

REVIEWER COMMENTS

Reviewer #1 (Remarks to the Author):

It is my opinion that the authors have constructively addressed the major concerns of both of the reviewers in this revised manuscript; I recommend acceptance for publication, although I have two minor issues with citations, one of which likely will confuse the reader on a small but significant scientific point. I know (from experience!) it is irritating for an author to receive requests for a second revision, but as I indicate below, the reader will be confused and inadequately informed if these two points are not addressed.

1. Page 15, lines12-13. The authors comment on the near-WT activity of the E486 missense variant expressed in intact cells:

“The E486K variant did not affect enzyme activity. . . in line with the previously demonstrated preserved enzyme activity in the absence of the ER retention peptide (30).”

The latter is not an appropriate citation. The quotation above strongly implies that there is no effect upon MINPP activity in intact cells upon solely removing the C-terminal ER retention sequence (which, incidentally, is a ‘sequence’, or a ‘motif’ or ‘signal’, not a peptide). If removal of ER retention sequence was all that was done in ref 30, the MINPP construct would still have been delivered to the ER. Instead, the MINPP construct described in ref 30 had been further engineered for removal of the N-terminal ER-targeting sequence. Consequently, the enzyme was shown to be mis-targeted to the cytosol.

That is, ref 30 describes an enzyme mislocalized to the cytoplasm (with increased access to substrate), whereas the author’s E486K construct would be delivered to the ER (where access to substrate is uncertain).

So, the author's data are certainly NOT "in line with previously demonstrated" observations in ref 30.

However, there is another publication in the literature, albeit with avian rather than human MINPP, that does indeed demonstrate there is no effect upon MINPP activity in intact cells upon solely removing the C-terminal ER retention sequence: PMID 16759730. The latter paper should be cited at this point in the ms, instead of ref 30.

Response: We thank Reviewer #1 for appreciating our previous work in fixing the major concerns, and also for this important new comment. We apologize for this error and have replaced the previous citation with the reference PMID 16759730. We have also replaced the word "peptide" by "signal" in page 15.

2. Page 5, 9-11 lines up. The authors emphasize novelty by claiming that "no Mendelian disorder has been shown to be caused by an imbalance in the cytosolic inositol polyphosphate pathway, with the exception of a single variant in a gene involved in the conversion of the pyrophosphates. . . " (my emphasis).

In fact, a second variant of the same gene has recently been described that also is associated with a separate Mendelian disorder of inositol phosphate turnover imbalance. See PMID 31852976. Because that latter study adds a little extra perspective to the novelty claim, it should also be cited.

Response: We now mention this new study (PMID 31852976).

Reviewer #2 (Remarks to the Author):

The points raised by the reviewer have been addressed. The inclusion of additional data has strengthened the manuscript

Response: We thank Reviewer #2 for considering the manuscript stronger and that the previous concerns are fixed.

Reviewer #3 (Remarks to the Author):

In this manuscript by Ucuncu and Rajamani et al., the authors describe mutations in the multiple inositol polyphosphate phosphatase 1 gene (MINPP1) as causative of Pontocerebellar Hypoplasia (PCH) in patients from three different families. In order to investigate how mutations in MINPP1 lead to this early-onset neurodegenerative syndrome, the authors investigate the accumulation of highly phosphorylated inositols (e.g. IP6) in HEK293 cells, patient fibroblasts, patient-derived iPSCs, genome-edited MINPP1-/- iPSCs as well as iPSC-differentiating neurons. The manuscript is well written and the data is presented in a clear fashion. There is substantial data pointing to dysregulation in inositol phosphate metabolism associated with mutations in MINPP1 in HEK393, iPSCs and immature neurons. However, it is important to highlight that, while the overall story is intriguing and the subject is novel and relevant, there are concerns that should be addressed regarding the stem cell model used in this work.

Major Issues:

1- To understand how mutations in MINPP1 could promote neurological dysfunctions observed in PCH patients, the authors attempted to differentiate iPSCs into cortical neurons. However, the differentiation was unsuccessful using a classic dual-SMAD inhibition protocol due to a lack of neural progenitor cells for the lines with MINPP1 mutation. The authors stated that this event "suggests a critical role for MINPP1 during neuroectodermal induction". This interpretation, thus, suggests that the mutations in MINPP1 studied in this experiment would model a possible neurodevelopmental disease as opposed to the neurodegenerative syndrome which is this type of PCH. This discrepancy between the data interpretation and the disease pathogenesis raises questions about how adequate is this model for this study and should be explored in the discussion.

Response: We would like to thank the reviewer for considering the work of interest, novel and relevant, and for these helpful comments. This is an important point, although complicated as PCH is generally seen as a disorder where neurodevelopment and neurodegeneration intersect prenatally. Thanks to the suggested characterization of cell deaths during neuronal differentiation, we obtained new results. We identified a significant ~ 2-fold increase in apoptosis in differentiating neurons with *MINPP1* mutations (see next point). Although this increased cell death likely contributes to the differentiation phenotype that we observed, it cannot explain the very significant increase in PAX6+ population in the *MINPP1* mutants as well as the potential role for this gene in neuroectodermal induction. Consequently, it suggests the co-occurrence of an inefficient, likely delayed, neuronal differentiation and the apoptosis of differentiating neurons. At this point, additional investigation beyond the scope of this paper will be needed to understand the link between these two aspects of the phenotype and their contributions to the disease pathogenesis. Overall, these results suggest that an iPSCs-based approach can provide relevant insights to better understand this gene defect and potentially other PCH sub-types. We have followed the Reviewer's suggestion and have modified the discussion based on the new results; we have also added "in vitro" to the sentence of the results section "suggests a critical role for *MINPP1* neuroectodermal induction", to limit the emphasis on this unsolved point.

2- The authors then use noggin as a single SMAD inhibitor and could only then successfully generate neurons. The comparison of mutant vs control lines showed a significant decrease in the percentage of TUJ1+ cells. To the same point of issue 1, given that the model is of a neurodegenerative disorder, it is important to understand if the progenitors (PAX6+) cells are not fully differentiating into neurons or if the cells differentiate until day 10+ and then degenerate. A cell death assay with time points post day 10 could help elucidate this issue.

Response: We have now assessed the apoptosis level using TUNEL assay at day 10 and 14. Interestingly, we observed a significant increase in apoptosis at day 10 of neuronal differentiation and that becomes more severe at day 14 (~ 2-fold; see new Fig. 3 C, D). By contrast, we did not observe significant differences in apoptosis levels among the undifferentiated iPSCs lines (new Fig. 3 D), suggesting a cell survival defect specific to the neuronal lineage and starting at early stage of neuronal differentiation.

3- The manuscript also lacks critical characterization of the iPSC differentiated neuronal cultures. Characterization of neural progenitor cells should be performed and should include Nestin in addition to PAX6. Fully differentiated neurons should also be characterized with at least MAP2 in addition to TUJ1. Most importantly, the experiments of IP6 accumulation was performed in d10 immature neurons, however, there is no representative image, in the main figures or supplementary, that show the neural cultures on that day. Additionally, it is only in the methods that the authors provide an explanation as to why d10 was chosen, this information should be present in the result section.

Response: We now provide the NESTIN staining for Day 14 differentiating neurons (see new Supplementary Fig.3 G). In our day 14 cell cultures, MAP2 positive neuronal population is detected but at very low level (Not shown) as it appears 3-4 weeks after rosette formation (PMID 16002783). Further, we now provide PAX6 and TUJ1 staining for Day 10 differentiating neurons. (Supplementary Fig. 3F). We also now indicate in the results section p.15 why day 10 was chosen.

Minor Issues:

1- The authors were careful not to overstate the neuronal cultures due to its immaturity, calling them "differentiating neurons" throughout the manuscript with the exception of the title on page 14: "Inositol polyphosphate metabolism is altered in HEK293, iPSCs and induced neurons mutated for *MINPP1*". "Induced neurons" should be substituted for immature neurons or differentiating neurons.

Response: We have now changed the word "induced neurons" to "differentiating neurons" for this title.

2- It is unclear why patient CERID-30-2 was chosen to be reprogrammed and this information should be present.

Response: We originally obtained reprogrammed iPSCs from CerID-30-1 and CerID-30-2 patients, but we identified a contamination with the cells derived from CerID-30-1, following the shipment of the lines from the Duke University iPSC core facility. Hence, we chose to work only with CerID-30-2 derived cells.

3- Moreover, the patient CERID-30-2 information describes MINPP1 and MTFMT as “genes with most likely pathogenic variants” however, MTFMT was not explored in the discussion.

Response: The MTFMT gene is mutated in a mitochondrial disorder that was excluded by clinical investigations. We have now added this information in the method part (supplementary note).

REVIEWERS' COMMENTS

Reviewer #3 (Remarks to the Author):

The major and minor issues raised by this reviewer have all been satisfactorily addressed. The addition of further characterization of iPSC and iPSC derived neurons, as well as, new data showing increased apoptosis in the affected culture significantly strengthen the manuscript. This reviewer recommends acceptance for publication.

Response to the reviewer's comments - Manuscript NCOMMS-19-20655C

REVIEWERS' COMMENTS

Reviewer #3 (Remarks to the Author):

The major and minor issues raised by this reviewer have all been satisfactorily addressed. The addition of further characterization of iPSC and iPSC derived neurons, as well as, new data showing increased apoptosis in the affected culture significantly strengthen the manuscript. This reviewer recommends acceptance for publication.

We thank the Referee for appreciating the revision work that we have performed and for the recommendation for publication.